# The Radiative Forcing Model Intercomparison Project (RFMIP): Experimental Protocol for CMIP6

Robert Pincus[1,2], Piers M Forster[3], and Bjorn Stevens[4]

[1]Cooperative Institute for Research in Environmental Sciences, University of Colorado, Boulder CO 80309 USA
[2]NOAA Earth System Research Lab, Physical Sciences Division, Boulder CO 80305 USA
[3]Institute for Climate and Atmospheric Science, School of Earth and Environment, University of Leeds, Leeds, UK
[4]Max Planck Institute for Meteorology, Hamburg 20146, Germany

*Correspondence to:* Robert Pincus (Robert.Pincus@colorado.edu)

**Abstract.** The phrasing of the first of three questions motivating CMIP6 – "How does the Earth system respond to forcing?" – suggests that forcing is always well-known, yet the radiative forcing to which this question refers has historically been uncertain in coordinated experiments even as understanding of how best to infer radiative forcing has evolved. The Radiative Forcing Model Intercomparison Project endorsed by CMIP6 seeks to provide a foundation for answering the question through

three related activities: (i) accurate characterization of the effective radiative forcing relative to a near pre-industrial baseline, and careful diagnosis of the components of this forcing; (ii) assessment of the absolute accuracy of clear-sky radiative transfer parameterizations against reference models on the global scales relevant for climate modeling; and (iii) identification of robust model responses to tightly-specified aerosol radiative forcing from 1850 to present.

Complete characterization of effective radiative forcing can be accomplished with 180 years (Tier 1) of atmosphere-only

simulation using a sea-surface temperature and sea ice concentration climatology derived from the host model's pre-industrial control simulation. Assessment of parameterization error requires trivial amounts of computation but the development of small amounts of infrastructure: new, spectrally-detailed diagnostic output requested as two snapshots at present-day and preindustrial conditions, and results from the model's radiation code applied to specified atmospheric conditions. The search for robust responses to aerosol changes relies on the CMIP6 specification of anthropogenic aerosol properties; models using

this specification can contribute to RFMIP with no additional simulation, while those using a full aerosol model are requested to perform at least one, and up to four, 165-year coupled ocean-atmosphere simulations at Tier 1.

## 1   Evolving understanding of radiative forcing

Perturbations to the chemical or physical state of the climate system, including those caused by anthropogenic activities, can induce a radiative perturbation loosely called a *radiative forcing*. Projections of future changes involve estimating the

magnitude of future radiative forcing and the strength of climate system's response to that forcing. If the system's response can be adequately described by a single temperature $T$ (normally the global-mean surface temperature) then radiative forcing $F$ is related to the top-of-atmosphere energy imbalance $N$ (or equivalently, global ocean heat uptake) and the temperature change

$\Delta T$ as

$$F = N + \alpha \Delta T \tag{1}$$

where the constant of proportionality between temperature and radiative response $\alpha$ is the climate feedback parameter.

Much attention has been paid to the differing responses of climate models to applied physical or chemical perturbations, especially in coordinated experiments such as including previous phases of the Coupled Model Intercomparison Project (CMIP, see e.g. Taylor et al., 2012). Differences in response are normally interpreted as differences in climate feedbacks arising from model formulations. Indeed this interpretation underlies the question "How does the Earth system respond to forcing?," one of the central questions motivating the sixth phase of the CMIP (CMIP6; see Eyring et al., 2016). The formulation of this question presumes that the forcing to which the earth system, or a model representation of the system, is subject is well-known. But this is not true in practice: models participating in exercises like CMIP are subject to surprisingly large differences in radiative forcing (Andrews et al., 2012; Forster et al., 2013; Chung and Soden, 2015) even when the perturbations applied to the physical system are the same. The radiative forcing to which the Earth itself has been subject is also relatively uncertain (e.g. Skeie et al., 2011). Even the concept of radiative forcing continues to evolve (Sherwood et al., 2015) in a search for informative measures and precise methods for diagnosis. To answer questions about how the earth system responds to forcing it is first necessary both to understand the nature of radiative forcing and to quantify the forcing experienced by individual models.

Some differences arise because individual models produce a range of radiative changes for the same physical perturbation. Specifying atmospheric composition changes, as has been common in previous phases of CMIP, does not uniquely determine even the instantaneous change in radiative fluxes at the top of the atmosphere (the "instantaneous radiative forcing" or IRF, in the languague of Myhre et al. (2013)). This is partly because extinction by gases depends on the distributions of temperature and humidity, which vary across models, and partly because computationally-efficient parameterizations for radiative transfer will differ to varying degrees with respect to reference models. This non-uniqueness is most relevant to radiative forcing by greenhouse gases, changes in which are responsible for the largest forcing since pre-industrial times (Myhre et al., 2013).

Differences also arise because models make different choices with respect to important but uncertain or loosely-specified physical perturbations, especially aerosols (Shindell et al., 2013) which, after greenhouse gases, are thought to be responsible for the second largest source of anthropogenic radiative perturbations. In the previous phase of CMIP the breadth of direct aerosol impacts to top-of-atmosphere radiation were larger than that due to greenhouses gases even as the signal was $\sim$3 times smaller (e.g. Myhre et al., 2013, Fig. 8.16).

Equation 1 is a diagnostic framework, useful in interpreting observations and comprehensive models of the climate system. Experience with models (in which all terms can be determined precisely) suggests that IRF is not, in practice, related very closely to changes in surface temperature, a point highlighted by Hansen et al. (1997) but well-known for even longer. Far more useful in Eq. 1 is the *effective radiative forcing* (ERF) that accounts for *adjustments*, the component of climate response that does not depend on global-mean surface temperature (Sherwood et al., 2015). Many such adjustments, for example the reduction in oceanic subtropical boundary layer cloudiness due to increased downwelling longwave radiation from increased $CO_2$, occur much more rapidly than the time scale for warming (e.g. Kamae and Watanabe, 2012) leading to the terminology

"rapid adjustments." Rapid adjustments are generalizations of (and replace) the stratospheric adjustment (Hansen et al., 1997) that has historically been used to account for the impact of rapid stratospheric equilibration on top-of-atmosphere radiation fluxes. Accurate diagnosis of ERF requires custom model integrations, either using linear regression to diagnose $F$ and $\alpha$ (assumed constant) in Equation 1 from temporal variations in $N$ and $\Delta T$ following abruptly-applied changes in composition (Gregory et al., 2004), or by approximately suppressing $\Delta T$ by fixing sea surface temperatures and inferring effective radiative forcing from $N$ following Hansen et al. (2005) and Rotstayn and Penner (2001). The diagnosis of ERF from such simulations is simplified, however, because ERF is diagnosed from changes in top-of-atmosphere radiation. Unlike the radiative perturbation arising from a composition change, ERF depends on the fullness of system response, so its calculation is no longer an exercise in pure radiative transfer.

Better estimates of effective radiative forcing will refine understanding of how the earth system responds to forcing, but the potentially knotty relationships between radiative forcing and response suggest value in subjecting models to ERFs that are as similar as possible. In signal processing it is common, when looking for a signal amidst a noisy background, to reduce the noise as close to the source as possible. In the context of ERF the largest source of variability is the treatment of atmospheric aerosol. RFMIP therefore includes coupled atmosphere-ocean simulations in which aerosol effective radiative forcing over the historical period is prescribed as much as possible, by analogy to protocols in which greenhouse gas concentrations over time are similarly specified. This is not to diminish the true uncertainty in historical concentration of anthropogenic aerosols but to ascertain what model responses robustly arise from a plausible historical aerosol radiative forcing.

The Radiative Forcing Model Intercomparision Project, RFMIP, seeks to provide a foundation for answering one of the guiding question of CMIP6, namely "how does the earth system respond to forcing?" This will be accomplished by

1. accurately characterizing the effective radiative forcing relative to a near pre-industrial baseline, and understanding the components of this forcing,

2. assessing the absolute accuracy of clear-sky radiative transfer parameterizations on the global scales relevant for climate modeling, and

3. identifying robust responses of comprehensive models to a specified aerosol radiative forcing over the period of instrumental measurements, i.e., 1850 to present.

This paper describes each of these efforts in greater detail, including the contributions requested from participating modeling centers, reference calculations to be undertaken as part of RFMIP, and planned analyses. The simulations are summarized in tables in Appendix A.

RFMIP follows CMIP6 protocols, so that "present-day" is interpreted as the year 2014 and "greenhouse gases" refer to those specified by Meinshausen et al. (2016), i.e. $CO_2$, $CH_4$, $N_2O$, and a long list of halocarbons. Ozone concentrations are specified separately from greenhouse gases but in concert with aerosols. Here we provide brief summaries of requested output but the definitive, detailed, and still-evolving specification is documented in the CMIP6 data request available at https://earthsystemcog.org/projects/wip/CMIP6DataRequest.

## 2 Diagnosing effective radiative forcing

The concept of radiative forcing has evolved over time, as can be seen by comparing the discussions in Hansen et al. (1997) with those in Sherwood et al. (2015), which we follow here. Partly for this reason, and partly because the climate system response was considered the largest unknown, previous iterations of CMIP have emphasized model response without careful characterization of effective radiative forcing. This omission has made it challenging to understand the degree to which differences in model response arise purely from different feedbacks. In the previous phase of CMIP, for example, models exhibited a wide range of global-mean temperature changes over the Historical period (1860-2005 for CMIP5). These models were driven by the same concentration or emission changes changes, but for the reasons described above the same concentration timeseries applied to different models led to different temporal evolutions of ERF including rapid adjustments (Forster et al., 2013). The extent to which the varying responses in CMIP3 and CMIP5 Historical simulations were due to differences in effective radiative forcing among models, as opposed to differences in feedbacks, remains unknown.

Limited understanding of ERF has severely hampered progress in key areas of physical climate science, including understanding historical temporal and spatial variations in climate feedbacks (Armour et al., 2013; Rose et al., 2014; Andrews et al., 2015); attribution of aerosol and greenhouse gas signals from the historic record (Bindoff et al., 2013); diagnosis of equilibrium climate sensitivity from observed energy budget changes (Masters, 2013; Otto et al., 2013); diagnosing transient climate response from historic trends (Gregory and Forster, 2008; Storelvmo et al., 2016); understanding the causes of global and regional precipitation trends (Richardson et al., 2016); and understanding of decadal variations in surface temperature, including the recent "hiatus" in surface warming (Marotzke and Forster, 2015; Fyfe et al., 2016).

RFMIP will diagnose model ERF by suppressing response, i.e. specifying sea surface temperatures and sea ice concentrations (Hansen et al., 2005). The "fixed-SST" method has important advantages compared to regressions of top-of-atmosphere imbalance against surface temperature change (Gregory et al., 2004). The most important is better error characteristics (Forster et al., 2016): thirty years of simulation using only the atmospheric and land components of an earth system model can diagnose global ERF to better than $0.05$ W/m$^2$ standard error, such that a $2\times CO_2$ effective radiative forcing of $3.7$ W/m$^2$ is larger than its standard error over 70% of the globe. Achieving similarly small errors from regression requires ensembles of coupled model integrations and therefore many centuries of simulation. Using fixed SSTs also allows model groups to diagnose transient ERF while regressions are suitable only for diagnosing ERF from abrupt changes. Transient ERFs are of particular interest in Historical simulations.

### 2.1 Protocol: Effective radiative forcing

The protocol for RFMIP fixed-SST integrations is to use a model-specific monthly-averaged climatology of SST and sea-ice based on the model's preindustrial DECK integration (Eyring et al., 2016). Applying a climatology limits variability and improves the diagnoses of small ERF differences. The same climatology is to be used for all ERF integrations. We request that distributions from a monthly averaged climatology of SST and sea-ice fractional coverage covering the annual cycle be generated from at least a 30 year segment of a preindustrial control integration. These should be prescribed according to the

AMIP protocols, whereby interpolated daily data is generated preserving the prescribed monthly averaged fields. Because ERF is weakly dependent on background state (Forster et al., 2016) the exact choice of background SST and sea-ice has little impact on the effective radiative forcing estimated in the historic period and in future climates (see below). We hope that a simple approach will encourage model centers to participate.

Time-slice simulations (Table 1), in which forcing agents are held constant at present-day or $4 \times CO_2$ values, provide estimates of present-day and $4 \times CO_2$ ERF. Present-day estimates provide a direct comparison between the estimates of ERF in the model with other estimates e.g. in assessment reports (Myhre et al., 2013). Estimate of ERF will also let us understand basic aspects of each model's temperature and other climate responses in the Historical and $4 \times CO_2$ DECK simulations.

Transient simulations (Table 2), in which forcing agent concentrations evolve over time, are designed to give a complete
picture of the CMIP6 Historical transient ERF and prospective future radiative forcing. Transient ERFs will be computed by differencing top of atmosphere energy diagnostics from three ensemble members employing time varying forcing-agent changes with the energy budget diagnostics from the 30-year control simulation. These integrations will use the same prescribed preindustrial climatology of SST and sea-ice as in the time-slice ERF experiments. AerChemMIP employs a more complex method of prescribing SSTs and sea-ice that allows for base climate changes through time. Offline tests found that such
complexity was unnecessary as ERF was only weakly dependent on the base state with small differences in the future confined to sea-ice edges (Forster et al., 2016) . Therefore RFMIP adopts the same base climatology in all experiments for ease of implementation. Tests also found that the transient ERF fields suffer from year-to-year random noise, so ten-year averages of the three ensemble members are needed to quantify ERF to within 0.05 W/m$^2$ (Forster et al., 2016). The future scenario (SSP2.4.5) matches experimental protocols requested by the Decadal Climate Prediction Project (DCCMIP, Boer et al., 2016)
and the Detection and Attribution Model Intercomparison Project (DAMIP, Gillett et al., 2016). The full ERF history from these simulations will give a much better understanding of decadal variability in the models and will aid attribution studies.

Land-surface models including interactive vegetation, if available, should be applied as in other CMIP integrations. The main diagnostics are the top of atmosphere energy budget terms required to estimate ERF. Diagnostics of atmospheric state, including temperature, water vapor, cloud and aerosol information, are requested to allow for detailed diagnosis of rapid adjustments. A
few daily variables related to temperature and precipitation are requested in conjunction with DAMIP to help distinguish direct effects of external forcing and air-sea interaction effects on historical changes in extreme indices (e.g., extreme precipitation).

We urge all centers to participate in "RFMIP-light" by performing the Tier 1 simulations in Table 1 even if they participate in no other aspect of RFMIP. Knowing the present-day and $4 \times CO_2$ ERF will enable modeling centers to understand why their DECK and Historical simulations differ from those performed by other models. Having all modeling centers perform these is
important to understand outliers in the multi-model ensemble, allowing us to probe if outliers are caused by radiative forcing-related or feedback-related processes. Further, the transient simulations in Table 2 are important for understanding model decadal variability and transient variations in climate feedbacks. This will benefit both decadal projections and attribution.

## 2.2 Planned analyses: Effective radiative forcing

Global and regional effective radiative forcing will be diagnosed for each model participating in RFMIP by differencing top-of-atmosphere radiative fluxes from the experiment with those from the preindustrial control simulation. RFMIP will characterize present day, historical and future ERF for the main radiative forcing groups ($CO_2$, all anthropogenic forcings, land-use, and aerosol and ozone changes taken together; see Tables 1 and 2). Aerosol and ozone changes are investigated together to allow participation from both concentration-driven and emission-driven models, as emissions of NOx, for example, can drive both ozone and aerosol changes. The complimentary Aerosols Chemistry Model Intercomparison Project (AerChemMIP, Collins et al., 2016) ERF simulations adopt the same calculation methodology as RFMIP for Tier 1 experiments. AerChemMIP targets interactive chemistry models and extends RFMIP to allow the community to further decompose present day aerosol and non-$CO_2$ ERFs into a more finely-delineated set of ERFs for different sets of precursor emissions.

Regional patterns of ERF will be compared across the models. This will aid the understanding of regional differences in climate response including an investigation of spatial variation in climate feedbacks.

The rapid adjustment component of effective radiative forcing will also be investigated. Rapid adjustments associated with aerosol-cloud-interaction are the major contributor to the negative aerosol ERF, and quantifying these effects has been a focus of much previous work (Boucher et al., 2013). Rapid adjustments are also important for many forcing agents including $CO_2$ (Sherwood et al., 2015). RFMIP requests joint histograms of cloud optical thickness and cloud top pressure from the "ISCCP simulator" (Klein and Jakob, 1999; Webb et al., 2001), part of the CFMIP Observation Simulator Package (Bodas-Salcedo et al., 2011) providing specialized diagnostics for the Cloud Feedback Model Intercomparison Project (CFMIP, see Webb et al., 2016). These will be used to estimate rapid adjustments by clouds using radiative kernels (Zelinka et al., 2012, 2014) that map changes in cloud properties into top-of-atmosphere radiative flux perturbations. Where these diagnostics are not available the approximate partial radiative perturbation methodology of Taylor et al. (2007) will be applied to clear and all-sky components of shortwave (SW) radiative fluxes, to estimate the rapid adjustments due to cloud changes. Non-cloud radiative kernels (Soden et al., 2008) will also be applied to standard diagnostics of water vapor and temperature to estimate IRF as well as stratospheric and tropospheric adjustments (Zhang and Huang, 2014; Chung and Soden, 2015).

These analyses will comprehensively characterize ERF in the each model participating in RFMIP. Radiative kernel diagnostics will enable us to develop understanding of rapid adjustment processes. We will test the kernel approach by comparing pre-adjustment radiative perturbations at the top-of-atmosphere estimated from radiative kernels with the best estimate of those perturbations from the line-by-line radiative transfer models according to the experimental design outlined in Section 3. In the same spirit we are interested in comparing radiative perturbations and cloud adjustments estimated from radiative kernels with those explicitly calculated in models employing the "triple radiation call" approach of Ghan (2013) to diagnose instantaneous radiative forcing and cloud adjustments. As this method is time-consuming and not implemented by all models we do not include this request as part of the protocol but models implementing triple radiation calls are encouraged to contact us.

## 3 Assessing parameterization error in clear-sky radiative forcing

One of the causes for model differences in effective radiative forcing for the same physical perturbation is error in radiative transfer parameterizations. This is somewhat surprising: radiative transfer is unique among the processes parameterized in atmospheric models because there is so little fundamental uncertainty. Line-by-line models can map atmospheric conditions and gas concentrations to extinction with very high accuracy and at very high spectral resolution. Transport algorithms, given enough computing resources, can compute fluxes to a precision limited primarily by uncertainty in inputs. But this deep knowledge is not completely represented in climate models. Parameterizations strike a practical compromise between accuracy and computational cost and might be expected to have some error even under the best of circumstances. More subtly, parameterizations require so much effort to develop and maintain that they can lag behind current spectroscopic knowledge, especially for solar radiation where new absorption features continue to be identified (Rothman et al., 2013). These errors have been apparent in previous assessments of radiative transfer parameterizations for both gaseous absorption (Ellingson and Fouquart, 1991; Collins et al., 2006; Oreopoulos et al., 2012; Pincus et al., 2015) and aerosols (Randles et al., 2013).

RFMIP will assess parameterization error in instantaneous radiative perturbations due to both greenhouse gases and due to aerosols. The assessments are independent and take quite different approaches but will both highlight global- and regional-mean errors.

Despite the important roles of clouds in modulating effective radiative forcing RFMIP focuses on parameterization error in cloud-free skies. This is partly because errors in clear skies are always present and may affect e.g. surface fluxes even when the top-of-atmosphere impact is masked by clouds, and partly because inter-model differences in the spatial and temporal distribution of clouds, which arise from complicated interactions between parameterizations and circulation, are likely to have a much larger impact on estimates of radiative forcing than are parameterization errors.

### 3.1 Protocol: Parameterization error

Assessments of radiative transfer parameterizations rely on computationally-expensive reference models. This has historically meant that only a few atmospheric conditions are considered, making it difficult to infer the error in global-mean radiative perturbations (Pincus et al., 2015) or the flux pairs which underlie them. The narrow range of conditions has also obscured some crucial differences between parameterizations, most notably the widely-varying sensitivity of shortwave absorption to water vapor that causes much of the variability in hydrologic sensitivity among climate models (Fildier and Collins, 2015; DeAngelis et al., 2015).

#### 3.1.1 Assessing accuracy in the treatment of greenhouse gases

RFMIP is developing a compact (roughly 100) sample of atmospheric conditions (profiles of pressure, temperature, humidity, greenhouse gas concentrations, surface properties) and radiative transfer boundary conditions (solar geometry and solar constant) that, when weighted appropriately, can be used to estimate time-averaged global-mean fluxes. (Sampling approaches are common in remote sensing problems; see for example Garand et al., 2001). Present-day atmospheric and surface conditions

are sampled from reanalysis while greenhouse gas concentrations follow the CMIP6 protocol, using 2014 values provided by Meinshausen et al. (2016). Aerosols are not included. Perturbations to these states allow for the calculation of IRF as the difference in flux between a perturbed state and present-day conditions and concentrations. Some perturbed states (see Table 3) represent changes in conditions tied to CMIP DECK or Historical simulations. The more idealized perturbations described in Table 4 are aimed at exposing model errors with global impacts, especially in present-day radiative forcing by specific greenhouse gases, while the three experiments in the table assess radiative transfer performance in conditions far from present-day. This set of conditions will be distributed on the Earth System Grid as a single file.

The sample is constructed to minimize the sampling error in annual-mean, present-day, clear-sky, aerosol-free radiative perturbation by greenhouse gases (i.e. the difference in top-of-atmosphere fluxes using present-day and pre-industrial gas concentrations). The sampling error, even with as few as 50 distinct conditions, is several orders of magnitude smaller than the flux perturbation itself. Sampling errors for other composition changes are larger but still small relative to the change in flux. Further details on the selection of these columns will be reported separately.

Modeling centers are asked to compute broadband (spectrally-integrated) fluxes for the full range of conditions and all perturbations using off-line versions of their radiative transfer parameterizations (or using any work flow that computes fluxes as the host model does using precisely the specified conditions). Modeling centers are asked to use the vertical grid provided and to omit aerosols. The representation of greenhouse gases, and particularly the choice of using a subset of gases or one of the equivalent concentrations provided by Meinshausen et al. (2016), should follow that used in other integrations made for CMIP6 and related activities.

Results from one or more reference models will also be made available on the ESG, as discussed in section 3.2.

### 3.1.2 Assessing accuracy in the treatment of aerosols

The assessment of aerosol instantaneous clear-sky (direct) radiative perturbations seeks to determine parameterization error "in the wild", i.e. under climatological conditions specific to each model. The effort is diagnostic: we request from modeling centers climate model estimates of clear-sky IRF due to aerosols and the detailed optical properties necessary to reconstruct this estimate, including instantaneous four-dimensional fields of spectrally-resolved surface albedo, aerosol extinction, single-scattering albedo, and asymmetry parameter on the models native atmospheric grid and using the native spectral discretization. The request is limited to solar radiation, and to a single day in the pre-industrial and present-day epoch taken from the model's CMIP6 Historical simulations. Participation involves no additional simulation but does require producing outputs new to CMIP that include a spectral dimension.

### 3.2 Planned analyses and supporting calculations: Parameterization error

For each calculation requested from modeling centers RFMIP will obtain matching calculations from one or more line-by-line reference radiative transfer models, allowing the accuracy of parameterization estimates of flux and radiative forcing to be assessed. One set of such calculations will be performed with a version of the LBLRTM radiative transfer model (Clough et al., 2005) updated to reflect recent changes to the HITRAN spectroscopic database (Rothman et al., 2013). Many of the reference

models participating in the intercomparison exercise described by Pincus et al. (2015) have indicated that they will also provide analogous results. Reference results will be provided for the sets of atmospheric conditions described in the last section – of order 2000 profiles for all perturbations. We anticipate that reference models may also be used to assess the impact of choices made in the CMIP6 specification for greenhouse gas concentrations (Meinshausen et al., 2016) including the use of equivalent concentrations to reduce the number of greenhouse gases considered, the neglect of species like CO that are not well mixed, and the specification of latitudinal and vertically-varying concentrations for well-mixed gases.

The diagnostic request for aerosol instantaneous radiative perturbation is substantially larger. Each model uses its own ambient atmospheric conditions, with order 65000 columns per time step for a 1 degree climate model. Eight 3-hourly time steps are requested for present-day and pre-industrial conditions. Reference calculations will be some combination of line-by-line modeling at reduced spectral resolution (though still much finer than in broad bands used in parameterizations) and subsets of columns sampled from each model to optimally represent present-day radiative forcing by aerosols.

## 4 Seeking robust signatures of aerosol radiative forcing

The calculations described in Section 3 are aimed at quantifying the degree to which parameterization weaknesses and errors impact estimates of radiative forcing – that is, the degree to which approximations and errors in radiative transfer parameterizations, as assessed against reference models, drives differences in radiative changes when the physical perturbation is fully specified. Although RFMIP seeks to understand this error for anthropogenic aerosols it is clear that the largest contributor to model differences in aerosol radiative forcing arises from distributions of anthropogenic aerosols (themselves the result of different prescriptions of aerosol precursors and processes), further modulated by varying distributions of clouds (Penner et al., 2006; Stier et al., 2013). As a result the temporal and spatial distribution of ERF caused by anthropogenic aerosols varies widely, greatly hindering attempts to identify and explain robust responses to aerosol perturbations including how anthropogenic aerosols affected twentieth century climate.

In the 20th century sulfate is thought to have contributed substantially to the net effective radiative forcing, although the magnitude and mechanisms are disputed (Stevens, 2015). What is not disputed is that precursor $SO_2$ emissions increased greatly, and that these emissions were concentrated over a relatively small portion of the planet. Consistent with other studies, Carslaw et al. (2013) estimate that $SO_2$ emissions, to which the dominant component of the aerosol contribution to ERF are attributed, increased three-fold through the first hundred years of industrialization. Smith et al. (2011) pinpoint these changes to changes in the North Atlantic sector – a region covering about a tenth of Earth's surface. Beginning in the 1970s air quality controls began to reduce emissions in Western Europe and North America. Present Western European emissions are now estimated to be a fifth, and North American emissions a half, of what they were in the early 1970s. As emissions over the Atlantic sector declined, emissions over South and East Asia increased so that globally anthropogenic $SO_2$ emissions remained roughly constant. The short life-time of sulfate implies that the regional concentration of emissions leads to strong regionality in radiative forcing. To the extent that sulfate forcing is important globally, then, regional signals should be readily identifiable and may help bound the overall radiative forcing attributable to anthropogenic $SO_2$ emissions.

One way to estimate the ERF from these changes to atmospheric composition is to calculate it from first principles i.e. from emissions information and chemical modeling. This approach is used increasingly frequently in earth system models but has so far led to wide disagreement in estimates of anthropogenic aerosol burden (Shindell et al., 2013) and aerosol ERF (cf Fig. 7.18 in Boucher et al., 2013). This disagreement is unsurprising: understanding of aerosol chemistry and physics is far from complete, and the ability to implement existing understanding is limited both by poor understanding of past emissions of aerosol and their precursors (Carslaw et al., 2013) and by incomplete understanding of aerosol interactions with other components of the climate system, especially clouds and precipitation (Stevens and Feingold, 2009; Bony et al., 2015).

Thus, beyond agreement that temporal and spatial changes in aerosols have been large, there is little consensus as to how such changes have influenced the twentieth century climate beyond reducing global-mean temperature by some indeterminate amount. Yet the response of the climate system to historical emissions of aerosols might offer the best chance of bounding aerosol ERF (Stevens, 2015). The strong warming in the first half of the century, a period when $CO_2$ concentrations rose only modestly, is difficult to reconcile with understanding of natural variability and a large (more negative than $-1 \, \mathrm{W/m^2}$) aerosol radiative forcing. This argument depends on the extent to which the climate response to a localized aerosol forcing is itself more localized than the response to a globally-distributed greenhouse gas forcing. In particular, if the northern hemisphere is subject to a net negative radiative forcing, but the global radiative forcing is slightly positive, is it reasonable to expect global warming that is northern-hemisphere amplified?

To answer these and similar questions it would be helpful to better understand how the climate system responds to a given aerosol perturbation in the presence of other physical perturbations such as increasing greenhouse gas concentrations. By more tightly constraining the pattern of the aerosol effective radiative forcing across models it should be easier to identify a clear response of the climate system to the imposed aerosol perturbations. To the extent that clear responses can be identified, they may be combined with formal methods of detection and attribution (e.g., Stott et al., 2010) to also estimate the magnitude of the radiative forcing.

The desire for a uniform, easily controlled and implemented representation of anthropogenic aerosol perturbations motivated the development of a semi-analytic representation of the distribution of anthropogenic aerosol-radiative and cloud-active properties over the full historical record. MACv2-SP (Stevens et al., 2016) specifies only the anthropogenic perturbation to the atmospheric aerosol and describes this perturbation directly and so does not interfere with the model development processes or tuning of the controlled coupled climate. The climatology prescribes the four dimensional distribution of anthropogenic aerosol radiative properties needed in two-stream radiative transfer calculations, i.e., the wavelength dependent aerosol optical depth, single-scattering albedo, and asymmetry factor. The influence of anthropogenic aerosol on clouds is specified as a multiplicative factor applied to the cloud droplet number concentrations used to calculate cloud droplet effective radius and hence cloud optical properties. Some models have experimented with representing a variety of more speculative aerosol-cloud interactions, for example increased cloudiness caused by changes to precipitation (Albrecht, 1989) that arise from aerosol perturbations. Given an increasing body of evidence (cf Christensen and Stephens, 2011; Boucher et al., 2013; Seifert et al., 2015; Haywood et al., 2016) calling these descriptions into question, MACv2-SP does not incorporate such effects.

The specification for MACv2-SP is described in detail by Stevens et al. (2016). In a version of the Max Planck Institute Earth System Model (MPI-ESM, see Giorgetta et al., 2013; Stevens et al., 2013) using an updated atmospheric component the clear-sky ERF for this aerosol description is -0.8 $W/m^2$, evaluated over the time period 2000-2011 using aerosols at 2005 values. The corresponding all-sky ERF is -0.685 $W/m^2$. In other models the latter value, especially, will depend on the model's distribution of cloudiness.

## 4.1 Protocol: Specified aerosol forcing

Simulations using MACv2-SP to describe the anthropogenic perturbation to the control background over the historical period (1850-2014) form the basis for the Specified Aerosol (SpAer) component of RFMIP. The simulations, described fully below and summarized in Table 5, repeat either DECK or other RFMIP simulations.

The recommendation for CMIP6 is that models using prescribed aerosol for the Historical simulations use the MACv2-SP specification (Stevens et al., 2016). Models using MACv2-SP to describe anthropogenic aerosols can participate in RFMIP-SpAer without additional effort by submitting the corresponding DECK or RFMIP simulation as part of RFMIP. Additional simulations beyond those needed to participate in CMIP6 or other components of RFMIP are only necessary if a modeling center *does not* adopt the MACv2-SP as their default aerosol prescription.

### 4.1.1 Tier 1 Simulation: hist-spAerO3-all

For this component of RFMIP only a single Tier 1 simulation with experiment_id *hist-spAerO3-all*, is requested. This simulation replicates the CMIP6-Historical simulation but using the MACv2-SP aerosol (Stevens et al., 2016) as the description of the anthropogenic aerosol forcing for models which use other representations for the CMIP6-Historical submission. A single ensemble member is required, but if it is intended to use this simulation also for DAMIP an ensemble size of four members is required.

### 4.1.2 Tier 2 Simulations

Tier 2 simulations are designed to augment the value of the Tier 1 simulations by making them useful for detection and attribution and to improve the diagnosis of radiative forcing. They either replicate simulations requested within DAMIP or within the ERF component of RFMIP. Again, they arise as an additional experimental request only for models that chose not to use MACv2-SP for their default description of the anthropogenic aerosol forcing.

**hist-spAerO3-aer:** This simulation is analogous to the Tier 1 simulation *hist-spAerO3-all*, except that the only time-varying forcing that is to be specified is that associated with the anthropogenic aerosol through the prescription of MACv2-SP. Volcanoes, solar variability and other non-aerosol forcings (both natural and anthropogenic) are to be omitted. Like *hist-spAerO3-all* it should use the full coupled (ocean-atmosphere) model and simulate the period between 1850 through 2014. For those models that adopt MACv2-SP as their default aerosol prescription it can replace the DAMIP aerosol-only simulation to satisfy the DAMIP protocol. Hence this additional simulation should only be performed for models wishing to contribute to DAMIP, and

in this case the Historical Natural Simulations must, through DAMIP, also be performed, i.e. historical simulations with only Natural Forcing.

**piClim-spAerO3-anthro:** This atmosphere-only simulation mimics the RFMIP *piClim-anthro* simulation described in Table 2 but using the MACv2-SP prescription of the anthropogenic aerosol as the aerosol component of the anthropogenic forcing. For what *piClim-anthro* describes as the "present day" aerosol, MACv2-SP provides a special description which averages aerosol properties for the period between 1985 and 2005. This is the first of four simulations intended to diagnose the ERF of aerosols and other anthropogenic perturbations. The first two diagnose ERF at present-day.

**piClim-spAerO3-AerO3:** This atmosphere-only simulation mimics the RFMIP *piClim-AerO3* simulation but using the MACv2-SP prescription of the anthropogenic aerosol as the aerosol component of the anthropogenic forcing. For what *piClim-AerO3* describes as the "present day" aerosol, MACv2-SP provides a special description which averages aerosol properties for the period between 1985 and 2005.

**piClim-spAerO3-histall:** This atmosphere-only simulation mimics the RFMIP *piClim-histall* simulation but using the MACv2-SP prescription of the anthropogenic aerosol as the aerosol component of the anthropogenic forcing. This is the first of two simulations aimed at diagnosing transient ERF in the presence of prescribed aerosols.

**piClim-spAerO3-histaer:** This atmosphere-only simulation mimics the RFMIP *piClim-histaer* simulation but using the MACv2-SP prescription of the anthropogenic aerosol as the aerosol component of the anthropogenic forcing.

### 4.2 Planned analyses: Aerosol forcing

Because the experimental design mimics that of the ERF component of RFMIP as well as allows for participation in DAMIP through a prescribed aerosol forcing, analysis will follow identically what is proposed for these families of simulations. In particular the SpAer-All experiments are planned for incorporation in formal detection and attribution studies to assess the magnitude of aerosol forcing.

The historical simulations based on MACv2-SP will be analyzed, also in cooperation with DAMIP, to test the hypothesis by Stevens (2015) that the observed northern-hemispheric warming is inconsistent with an aerosol radiative forcing more negative than about $-1\mathrm{W/m^2}$. The Tier 1 experiment hist-spAerO3-all will also be used to identify robust responses to an aerosol forcing. For example the pattern, or lack thereof, of the response across the multi-model ensemble may be helpful to advancing our understanding of the extent to which aerosol forcing underlies the warming hole in the east-central United States (Leibensperger et al., 2012), shifts in the tropical convergence zones (Bollasina et al., 2011), or phasing of Atlantic (Booth et al., 2012) and Pacific (Meehl et al., 2009; Smith et al., 2016) decadal variability. Tier 2 experiments are primarily concerned with allowing analysis already planned to also be performed for models with the MACv2-SP aerosol, for instance piClim-spAerO3-anthro will be used to characterize how different the ERF is for an identical specification of aerosol optical and cloud active properties, and to what extent these differences arise from differences in the adjustments or in the instantaneous radiative perturbations being differently masked by atmospheric properties.

## 5   Summary

CMIP6 addresses three broad questions: (i) how does the Earth system respond to forcing?, (ii) what are the origins and consequences of systematic model biases?, and (iii) how can we assess future climate changes given climate variability, limited predictability, and uncertainties in scenarios? (Eyring et al., 2016). As we have noted, results from all phases of RFMIP will be central in addressing question (i) both by better charactering the ERF relevant to each model's Historical simulation (in RFMIP-ERF) and by examining the response of those same models to far more tightly-constrained ERF due to aerosols (RFMIP-SpAer). RFMIP will contribute valuable information on model biases (question ii) through the assessment of radiative transfer parameterizations on global scales (RFMIP-IRF) and help reduce, in a small way, the uncertainty in scenarios caused by error in the translation of gas concentrations to radiative flux perturbations.

RFMIP also supports elements of the World Climate Research Program's Grand Science Challenges. Links are especially strong to the effort on Clouds, Circulation, and Climate Sensitivity (Bony et al., 2015, with which BS and RP are involved) through a shared interest in cloud adjustments, for which the ISCCP simulator diagnostic information requested in section 2.2 will be quite useful. Many of the challenges have strong regional aspects that may benefit from the RFMIP-SpAer simulations in which the regional forcing is constrained to be more similar across models than has been true to date.

RFMIP also offers a chance to explore methods for model development and experimental protocols. The assessment of radiative transfer parameterizations has a 25+ year history but such assessments have often been performed on a narrow range of idealized conditions, obscuring their relevance to climate model response until underlying errors become evident in important aspects of model response (e.g. Fildier and Collins, 2015; DeAngelis et al., 2015). By identifying a tractably-sized but globally-representative set of conditions we hope to enable routine testing of parameterizations stringent enough to identify errors during model development; these will provide a useful complement to observationally-constrained conditions (Oreopoulos et al., 2012) useful for testing reference models.

**Data availability**

All data requested by RFMIP will be distributed through the Earth System Grid Federation (ESGF) with digital object identifiers (DOIs) assigned, as will the inputs required for offline radiative transfer calculations described in Section 3 and results from reference models. It is the intent of RFMIP that this data be freely available; our expectation is that users of the data will give proper credit to the groups producing that data (i.e. by referencing the relevant DOIs) and generally comply with the recommendations of the WGCM Infrastructure Panel as described in their invited contribution to this Special Issue, including acknowledging CMIP6, the participating modelling groups, and the ESGF centres (see details on the CMIP Panel website at http://www.wcrp-climate.org/index.php/wgcm-cmip/about-cmip).

**Table 1.** Experiments for diagnosing effective radiative forcing at present-day and under $4\times CO_2$ conditions. These are atmosphere-only integrations with interactive vegetation using sea-surface temperatures and sea ice concentrations fixed at model-specific pre-industrial control climatology. All experiments are perturbations to RFMIP-ERF-PI-Cntrl and are requested at Tier 1.

| Experiment Title | CMIP6 Label (experiment_id) | Experiment Description | Years | Major Purposes |
|---|---|---|---|---|
| RFMIP-ERF-PI-Control | piClim-control | Pre-industrial conditions | 30 | Baseline for model-specific effective radiative forcing (ERF) calculations |
| RFMIP-ERF-Anthro | piClim-anthro | Present-day anthropogenic forcing (greenhouse gases, aerosols and land-use) | 30 | Quantify present-day total anthropogenic ERF |
| RFMIP-ERF-GHG | piClim-ghg | Present-day greenhouse gases | 30 | Quantify present-day ERF by greenhouse gases |
| RFMIP-ERF-AerO3 | piClim-aerO3 | Present-day aerosols and ozone | 30 | Quantify present-day ERF by aerosols and ozone |
| RFMIP-ERF-LU | piClim-lu | Present-day land use | 30 | Quantify present-day ERF by land use changes |
| RFMIP-ERF-4xCO2 | piClim-4xCO2 | $CO_2$ concentrations set to 4 times pre-industrial | 30 | Quantify ERF of $4\times CO_2$ |

**Table 2.** Experiments for diagnosing time-evolving effective radiative forcing. Three-member ensembles of atmosphere-only integrations interactive vegetation and using sea-surface temperatures and sea ice concentrations fixed at model-specific pre-industrial control climatology. Forcing post-2015 uses a scenario consistent with DCPP and DAMIP (SSP2-4.5)

| Experiment Title | CMIP6 Label (experiment_id) | Experiment Description | Start | End | Major Purposes |
|---|---|---|---|---|---|
| RFMIP-ERF-HistAll | piClim-histall | Time-varying forcing from all agents. | 1850 | 2100 | Diagnose transient ERF from all agents |
| RFMIP-ERF-HistNat | piClim-histnat | Time-varying ERF from volcanoes, solar (including spectral) variability | 1850 | 2100 | Diagnose transient natural ERF |
| RFMIP-ERF-HistGHG | piClim-histghg | Time-varying ERF by greenhouse gases | 1850 | 2100 | Diagnose transient ERF from greenhouse gases |
| RFMIP-ERF-HistAerO3 | piClim-histaerO3 | Time-varying ERF by aerosols | 1850 | 2100 | Diagnose transient ERF from aerosols |

**Table 3.** Sets of atmospheric conditions to be supplied by RFMIP for assessing parameterization error in clear-sky top-of-atmosphere flux changes. The entire set of conditions is described as CMIP Experiment RFMIP-IRF with CMIP6 Label (experiment_id) rad-irf

| Atmospheric conditions | Gas concentrations | Major Purpose | Relevant Experiment |
|---|---|---|---|
| Present-day | Present-day | Baseline | |
| Present-day | Pre-industrial | Present-day radiative forcing | Historical |
| Present-day | $4\times$ pre-industrial $CO_2$ | Radiative forcing from $4\times CO_2$ | abrupt4xCO2 |
| Present-day | "future" | Radiative forcing in future conditions | RCP8.5 at 2100 |

**Table 4.** Sets of atmospheric conditions to be supplied by RFMIP for assessing radiative forcing by specific agents and probing sources of parameterization error. The entire set of conditions is described as CMIP Experiment RFMIP-IRF with CMIP6 Label (experiment_id) rad-irf

| Atmospheric conditions | Gas concentrations | Major purpose |
|---|---|---|
| Present-day | Pre-industrial $CO_2\times0.50$ | Radiative forcing dependence on $CO_2$ |
| Present-day | Pre-industrial $CO_2\times2$ | Radiative forcing dependence on $CO_2$ |
| Present-day | Pre-industrial $CO_2\times3$ | Radiative forcing dependence on $CO_2$ |
| Present-day | Pre-industrial $CO_2\times8$ | Radiative forcing dependence on $CO_2$ |
| Present-day | Pre-industrial $CO_2$ | Present-day radiative forcing by $CO_2$ |
| Present-day | Pre-industrial $CH_4$ | Present-day radiative forcing by $CH_4$ |
| Present-day | Pre-industrial $N_2O$ | Present-day radiative forcing by $N_2O$ |
| Present-day | Pre-industrial $O_3$ | Present-day radiative forcing by $O_3$ |
| Present-day | Pre-industrial HFC | Present-day radiative forcing by hydroflourocarbons |
| Present-day +4K | Present-day | Assess error in temperature dependence |
| Present-day +4K | Present-day, water vapor increased to present-day relative humidity | Assess error in sensitivity to water vapor |
| Pre-industrial | Pre-industrial | Sensitivity of combined concentration/condition changes |
| "Future" | "Future" | Sensitivity of combined concentration/condition changes |
| Present-day | Last glacial maximum per PMIP | Assess errors under LGM conditions as per Kageyama et al. (2016) |

# Appendix A:  Summary of requested simulations and other calculations

*Acknowledgements.*  RP and PMF are financially supported by the Regional and Global Climate Modeling Program of the US Department of Energy Office of Environmental and Biological Sciences under grant DE-SC0012549. The protocol for RFMIP has benefited from conversations with T. Andrews, D. R. Feldman, E. J. Mlawer, L. Oreopoulos, and D. Paynter. W. D. Collins and V. Ramaswamy originated the idea

**Table 5.** RFMIP simulations with specified anthropogenic aerosols (SpAer). All simulations are all based on the MACv2-SP prescription of anthropogenic aerosol optical and cloud active properties. They are only to be performed to replicate other simulations in the DECK, within the ERF component of RFMIP, or within the Detection and Attribution Model Intercomparison Project (DAMIP) in the case when MACv2-SP is not used as the default aerosol climatology in the parent simulation.

| Experiment Title | Experiment_id | Tier | Period (or years) | Members | Parallel Experiment_id |
|---|---|---|---|---|---|
| RFMIP-SpAerO3-all | hist-spAerO3-all | 1 | 1850-2014 | 1 (4) | CMIP6-Historical |
| RFMIP-SpAerO3-aer | hist-spAerO3-aer | 2 | 1850-2014 | 4 | Historical-Aer (DAMIP) |
| RFMIP-SpAerO3-ERF-anthro | piClim-spAerO3-anthro | 2 | 30 | 1 (4) | piClim-anthro (RFMIP-ERF) |
| RFMIP-SpAerO3-ERF-aer | piClim-spAerO3-aerO3 | 2 | 30 | 1 (4) | piClim-aerO3 (RFMIP-ERF) |
| RFMIP-SpAerO3-ERF-histall | piClim-spAerO3-histall | 2 | 1850-2014 | 1 (4) | piClim-histall (RFMIP-ERF) |
| RFMIP-SpAerO3-ERF-histaer | piClim-spAerO3-histaer | 2 | 1850-2014 | 1 (4) | piClim-histaer (RFMIP-ERF) |

of assessing parameterization error in treatment of aerosols (section 3.1.2). We thank Daniel R. Feldman of Lawrence Berkeley National Lab for carefully assembling the novel data request for the assessment of aerosol radiative forcing.

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
