# Peer review of "The Radiative Forcing Model Intercomparison Project (RFMIP): Experimental Protocol for CMIP6"

_Geoscientific Model Development, 2016_

## Referee Comment (RC1) · K.P. Shine (Referee) · 27 May 2016

Review of The Radiative Forcing Model Intercomparison Project by R. Pincus et al. (doi: 10.5194/gmd-2016-88)

I regard this particular MIP as having high importance and thank/congratulate the author team for their leadership.

My comments are mostly rather minor, and will be easy to deal with.

2-(6-7): I agree with the "normally interpreted" here, but there have been important indications that diversity in the forcing is also responsible, most notably Chung and Soden (2015 doi:10.1088/1748-9326/10/7/074004) in the recent literature.

2-13: "Observational estimates of the radiative forcing ..." I didn't quite get this sentence. The reference to Skeie et al. seems strange given that in their paper "for short lived climate forcers, detailed chemical transport modelling and radiative transfer modelling using historical emission inventories is performed". I wondered if this was mixing up other work by the same author team on quasi-observationally based estimates of climate sensitivity, rather than forcing?

2-10: IRP – presumably this means "instantaneous radiative perturbation" but this isn't spelled out, nor is IRP used in some of the following paragraphs (it is given in full). I'm also not quite sure why IRP is "more precise" than IRF, but perhaps I miss something. (Note typo "language" on same line.)

2-22 and 2(32-33): The masking effect of clouds is also normally included in the IRP (and stratosphere-adjusted forcing) – I wasn't sure why it was just mentioned in the context of ERF. The difference with ERF is that the clouds can respond to the forcing.

6-5 "somewhat surprising" and 6-10 "might be expected to have some error" seem contradictory. I'm not surprised, by the way. We've even seen that different implementations of the same radiation code lead to different forcings (e.g Myhre et al. 2009, 10.1127/0941-2948/2009/0411). Perhaps this shows that codes should be tested in implementations as close to those used in the ESM as possible, and this is a point made by the Chung and Soden paper referred to above.

6-11 "current spectroscopic knowledge" – I am not sure what the evidence is that radiation parameterisation error is due, to a significant degree, to spectroscopic knowledge. For example, Kratz (10.1016/j.jqsrt.2007.10.010 – see especially his Table 6) shows rather small impacts of changing HITRAN database for any post-1990 data base, and indeed concludes "the line parameter updates to the HITRAN database are not a significant source for discrepancies in the radiative forcing calculations", and that is also my overall experience.

6(17-21) I slightly lost the plot here – firstly I am not sure what evidence there is that

cloud optical properties is "likely to have a larger impact" is (especially in the context of global mean forcing), and if it is the case, it seems to undermine the reason for focusing on clear skies.

6(25) "obscured" – I think the intermodel spread is pretty evident in Collins et al. (2006) where the tropopause standard deviation is almost as big as the mean; I think what has happened in more recent years is a realisation that this really matters for hydrologic(al) sensitivity.

7(2) I had a few comments on this table. One is that specification of "HFC", rather than a specific HFC, seems too vague to me, and I am not sure what will be learnt unless there is a tighter specification. Second the +4K experiment is well motivated, but not really elaborated on – I guess the focus is on the temperature dependence of transmittance, but this experiment might also have some Planck function dependencies in the results, which will be harder to tease apart. Finally note a typo for the ozone experiment, where it says O2 not O3.

7-23: Not entirely sure why LBLRTM is singled out here – it is, of course, a very important resource, but not the only reference code. Perhaps the reference to Pincus et al. (2015) suffices?

7-30 Perhaps the most major of my comments. I naturally assumed that the shortwave GHG forcing would be included for the greenhouse gases, but I realised at this point that while the insolation conditions are specified for aerosols, they are not specified for greenhouse gases. I would say that one drawback with earlier shortwave GHG comparisons is the simple (single zenith angle etc) specification of the shortwave parameters, and it would be advantageous to have a proper day/global aversge even if it is for a single day, as in the aerosol case. Could the authors at least clarify the situation regarding the GHG SW forcing?

8-3 "parameterization error increases model diversity" – isn't parameterization error a (and possibly the major) CAUSE of model diversity (unless all models have the same

error!).

8-9 This feels like sloganeering, and I haven't really got a clue what "the 20th century belonged to sulphate" means (there are many contenders to the accolade of "the 20th century belonged to ..."! I might vote for the Beatles), especially given the claim that a strong negative aerosol forcing is implausible and the statement that greenhouse gas forcing dominates at 2-23. Incidentally, is the reference to Carslaw et al., in the following sentence, the right one? There seems little or nothing on sulphate global optical depths in the cited paper.

8-13 "Starting in the mid-1970s" – maybe better "Since the mid-1970's"? I presume the stated changes (five-fold, factor of two etc) refer to the present day compared to mid-1970s?

8-29 "less negative" – isn't this "more negative"?

9-3 typo "response"

9-18 This section was written in a different way to the equivalent in Section 3. I could not see a reference to Table 5, and the section is a bit cluttered by the acronyms of the experiments, which are a bit opaque (to me). Perhaps this section could be restructured (not a big job) to do as was done in Section 3, and leave the acronyms to the table, but ensure the underlying motivation of experiments is spelled out.

9-31 "warming" – sorry if I am getting confused, but as I understand, RFMIP only provides evidence of "forcing" rather than "warming". Perhaps it is the coupling with DAMIP that is being referred to here?

10-2 "warming hole" – same comment as above. In fact the whole of this sentence, seems to go a bit beyond what RFMIP results could achieve (although they give important clues), so I wondered if this was really referring to DAMIP.

---

## Referee Comment (RC2) · J. Quaas (Referee) · 15 Jun 2016

Pincus, Forster and Stevens describe the concepts of the Radiative Forcing Model Intercomparison Project (RFMIP), intended as a contribution to the 6th Coupled Model Intercomparison Project (CMIP6). In my opinion, RFMIP is a key contribution to CMIP6 and an outstanding advance over the previous CMIP phases. It is of utmost value that for the historical period, but also for the future scenarios as well as for the idealised simulations (4xCO2) the effective radiative forcing (ERF) and its spatiotemporal distribution is quantified. This will help enormously to advance the interpretation of simulated responses, and of detecting and attributing climate change signals.

RFMIP proposes to do so in a series of three distinct sub-projects. The first one consists of conducting simulations with prescribed, pre-industrial distributions of sea surface temperatures (SST) and sea ice cover (SIC) that allow to diagnose the ERF for different combinations of forcing agents at different times. The second is split into two and intends to compare model-simulated radiative transfer results to reference computations, on the one hand for greenhouse gases (CO2?), where pre-selected thermodynamic profiles are used, and on the other hand for aerosols, where global snapshots from the contributing general circulation models (GCMs) are used. Finally, a time-varying climatology of the anthropogenic perturbation to aerosol radiative properties and cloud droplet number concentrations is used to consistently impose historical aerosol ERFs to all participating GCMs.

The concept in general is excellent, and will help the science a lot.

My main remark is that at several instances, more details would be useful to exactly define the RFMIP in order to very precisely describe the set-up to model centres.

I have a few more overarching suggestions, and several minor ones.

*General remarks*
- The paper only occasionally indicates which model diagnostics (output) is to be produced and analysed. One could only assume that the different contributions of top-of-atmosphere (also surface?) radiation flux densities (solar and terrestrial, all-sky and clear-sky) are to be determined. What about cloud quantities?
- When it comes to the aerosol ERF it has been shown that it is useful to on-line diagnose components of the forcing. This in particular involves a triple-call to the radiation, allowing to diagnose the radiative forcing due to aerosol-radiation- and due to aerosol-cloud interactions (Ghan, Atmos Chem Phys 2013; doi:10.5194/acp-13-9971-2013). Would it not be useful to request such a diagnostics?
- For the transient ERF, more specifications are necessary how to compute it. Are multiple ensemble members necessary? Or are time slices computed? Or else is a noisy signal accepted? What about the strong deviation from the pre-industrial base state at

least in the future scenarios at least in the Arctic?

- For the IRF study, the paper should be structured more clearly to clarify the two aspects (CO2 and aerosols) to this, which use two very distinct approaches.

- for the imposed aerosol forcing, it would be good to indicate what the implied ERF due to aerosol-radiation-interactions and what the ERF due to aerosol-cloud-interactions are for e.g. year 2011.

*Specific remarks*

page 1
l14 relies
l19 the "roughly" might merit a sentence of explanation, or a reference.
L22 "is related" → maybe "is approximately linearly related"
is it worth mentioning that Eq. (1) actually defines the radiative forcing, and that the usefulness of F is linked to the degree to which $\alpha$ is independent on the exact nature of the process that generates F, so that comparing different F is sufficient to predict different $\Delta T$ for a given $\alpha$ (i.e. given model)?

l17 "individual models" → "different models"?
L20: If sticking to the acronym "IRP" this should probably be spelled out "instantaneous radiative flux perturbation" (assuming the "R" represents "radiative")
l30: this sentence in my opinion is not understandable to somebody not experienced in the topic. The point is about efficacy, and so I'd propose to write "...suggests that IRP due to different forcing agents is not, in practice, a very good predictor for changes in surface temperature assuming constant climate feedback parameters, a point..."
l33: I'd suggest to be more specific: "...does not depend on global-mean surface temperature change (Sherwood et al., 2015)."

l2 "accurate diagnosis", or rather diagnosis at all?

L5 Reference Hansen et al. (2005): I think it would be appropriate to also cite Rotstayn and Penner (J Climate 2012, 14, 2960-2975) who introduced the concept several years before Hansen

l6 "fullness of the model response" to be precise, perhaps "fullness of the rapid model response"

page 4

l2 "concentration or emission changes", since at least some models computed carbon- and aerosol cycles interactively

l15 One would expect a reference corroborating the sentence, rather than – once more – introducing the regression concept (Gregory).

L19 The sentence is not straightforward to understand. For both methods, only a given perturbation can be diagnosed, and this can be done for either approach. L22 It would be useful to specify as clearly as possible the simulation. I'd suggest to clarify the following things: (i) it is one annual cycle consisting of 12 geographical distributions of sea surface temperatures and sea ice fractional coverage; (ii) from how many years should the fields be derived? Average over any 30 years of the piControl run? (iii) should the monthly means be linearly interpolated to the individual time step, or abruptly change on the first of each month, 0 UTC?

L26 "without compromising accuracy" - I don't understand what is meant in this context.

L27 "present-day" needs clarification. Is this year 2011 as in CMIP5, or 2005 as in CMIP3? It would also be good this time to have the date consistent with the end of the DECK historical simulation.

The Tier-2 experiments need to be motivated. Is a signal from 0.1 Aer really detectable over the noise in a 30-year fixed-SST framework? Why a factor of 0.1 and not a factor of 0.5? Or why not – probably even more useful in interpreting the historical simulations – a year 1985 simulation?

l2 Forster et al. (the cited JGR paper) say 30 years of integration time are necessary to characterize ERF. How is this done here? By 30 ensemble members and they reporting the forcing every year of the simulation? Or is a much greater transient uncertainty simply accepted if a single ensemble is run?

It seems Forster et al. (JGR) claim the base state does not matter much so a pre-industrial SST and SIC distribution is good enough. But is this not a main uncertainty when integrating into the future? How meaningful is a forcing diagnostics in the second half of the 21st century in the Arctic when pre-industrial sea ice cover is prescribed?

l14: "both greenhouse gases and aerosols": It is quite unclear from Table 3 how the greenhouse gases beyond $CO_2$, and aerosols are to be prescribed, in both the control and perturbed simulations. Only at some point in the text, it appears the atmospheric profiles should be not only cloud-free, but also aerosol-free. I didn't find any information about greenhouse gases beyond $CO_2$.

l1 Why not provide the selected profiles as supplementary material? It would be useful to specify in the paper the exact way the profiles are introduced. Is the vertical discretisation the model's one? Or is a common vertical grid chosen?
L4 "aerosol-free": What about greenhouse gases?
L8 "when finalized" why not put this up for review here as well?
L12: It would be good to make clear at the beginning of section 3 that there are two distinct and very different approaches to characterise greenhouse gas- and aerosol forcing. It would be also useful to split these two into to sub-sections, one on greenhouse gases, and one on aerosols.
L14: "radiative perturbations": does this imply a double radiation call, one with the current aerosol and a second one with all aerosol set to zero? The exact definition should be clarified, and also the necessary output. Is this requested for the top of the

atmosphere, or vertically resolved? Broadband in the solar spectrum, or spectrally resolved? Why not both, the clear-sky and the clean-sky flux?

L15: "surface albedo, aerosol" (with ",")

l17: It would be good to be exhaustive in the request list, i.e. to also include which other parameters (temperature, pressure, specific humidity? gases?) to provide.

L24: Maybe would it be possible to write why multiple reference models are advantageous? Is it expected that their results differ?

l6 Another substantial contributor is the diversity in simulated cloud distributions (Penner et al., Atmos Chem Phys 2006 doi:10.5194/acp-6-3391-2006; Stier et al., Atmos Chem Phys 2013, doi:10.5194/acp-13-3245-2013)

l9: 20th century: rather a balance of sulfate and greenhouse gases.

L12: To be more precise: the continents adjacent to the North Atlantic

l17: "commensurately large", or why "larger" (than the emission changes?)

l26: "reducing temperature" may be misunderstood (the GHG effect mostly is dominant), why not – pertinent to the paper – say that they introduce a negative effective radiative forcing?

l2 "perturbation" singular

l17 It would be useful to indicate the ERF of the MACv2-SP aerosol perturbation e.g. for year 2011.

l23 Why citing Eyring here?

L23: The sentence in brackets is difficult to understand and should be explained better.

l33: But does the MACv2-SP provide a forcing more negative than -1 Wm-2 in 2011?

l10 "addresses" (present tense)

[Figure]

l21: "quite useful" for what?

Table 1
"Present-day" is not specified. Is this intended?

CMIP6 label/experiment id: can the dashes be omitted? I think having just words helps in some scripts, and this was common practice in CMIP5.

RFMIP-ERF-GHG Experiment description: It should be made very clear whether ozone (tropospheric? Stratospheric?) is considered a (greenhouse) "gas". Can one not be specific by writing "CO2, CH4, N2O, Halocarbons, and O3 and CO precursor gases?

RFMIP-ERF-AerO3: Experiment description: How is O3 perturbed for models that include atmospheric chemistry, i.e. that require emission, rather than concentration, perturbations? Is this tropospheric ozone only?

RFMIP-ERF-LU Experiment description: gases or land-use?

RFMIP-ERF-4xCO2: Does this require another control simulation for models with an interactive carbon cycle (namely a simulation with pre-industrial CO2 concentrations, rather than emissions)?

RFMIP-ERF-AerO3x01: It should be clarified that the anthropogenic O3 and aerosol concentrations, or ozone precursor and aerosol (precursor) emissions should be scaled by a factor of 0.1. (same for RFMIP-ERF-AerO3x2)

Table 2

RFMIP-ERF-HistNat Experiment description: I think "etc." is very bad to use in such a protocol. It is necessary to very precisely say what should be varied. What could and should be thought of besides solar activity and volcanic eruptions?

RFMIP-ERF-HistAer Experiment description: What about ozone?

Table 3. "in forcing for ." for what?

---

## Author Comment (AC1) · 16 Jun 2016

Dear Keith -

Thanks very much for these helpful comments. Our responses are below.

*2-(6-7): I agree with the "normally interpreted" here, but there have been important indications that diversity in the forcing is also responsible . . .*

Yes, and indeed the point of RFMIP is that some of the diversity in response is likely to due diversity in forcing. In revisions will make this idea more explicit and cite the Chung and Soden paper in support of the point.

[Figure]

*2-13: "Observational estimates of the radiative forcing . . . " I didn't quite get this sentence. The reference to Skeie et al. seems strange . . .*

It's true that estimates of radiative forcing as in the Skeie paper necessarily involve rather a lot of modeling, both with respect to radiative transfer and to composition, so that calling that work an "observational estimate" is overstating the case. We'll choose a more nuanced way to make this point in the revisions.

*2-10: IRP – presumably this means "instantaneous radiative perturbation" but this isn't spelled out, nor is IRP used in some of the following paragraphs (it is given in full). I'm also not quite sure why IRP is "more precise" than IRF, but perhaps I miss something.*

We introduced the IRP nomenclature late in the writing of the manuscript. We are trying find language that clearly distinguishes between theoretical changes to fluxes, i.e. IRF or IRP, that are never actually realized, and the actual change in energy balance to which the climate system responds, i.e. ERF including adjustments. But we clearly didn't make this argument convincingly enough (reviewer 2, Johannes Quaas, is similarly confused) so we'll revisit this carefully in the revisions.

*2-22 and 2(32-33): The masking effect of clouds is also normally included in the IRP (and stratosphere-adjusted forcing) – I wasn't sure why it was just mentioned in the context of ERF. The difference with ERF is that the clouds can respond to the forcing.*

Good point – we were trying to emphasize that even clear-sky instantaneous flux changes might differ among models for the same change in composition, but we'll mention cloud masking at 2-22 as well.

*6-5 "somewhat surprising" and 6-10 "might be expected to have some error" seem contradictory. . . . Perhaps this shows that codes should be tested in implementations as close to those used in the ESM as possible, and this is a point made by the Chung and Soden paper referred to above.*

We are trying to distinguish between true uncertainty, which is very small, and the

approximation/parameterization errors we aim to assess.

We debated asking people to test the radiation models as you describe, basically as they are implemented in the host model. The drawback is that then each model will work on its own vertical grid, so different models won't be solving precisely the same problem.

*6-11 "current spectroscopic knowledge" – I am not sure what the evidence is that radiation parameterisation error is due, to a significant degree, to spectroscopic knowledge. For example, Kratz (10.1016/j.jqsrt.2007.10.010 – see especially his Table 6) shows rather small impacts of changing HITRAN database for any post-1990 data base . . . and that is also my overall experience.*

It's true that spectroscopic knowledge in the longwave, as reflected in the HITRAN databases, has not changed much in the last 25 years, at least for "normal" atmospheric conditions. But the number of known absorption lines in the shortwave continues to increase, as the HIRTRAN compilers themselves note (10.1016/j.jqsrt.2013.07.002), and it's in the shortwave where parameterizations vary most widely, especially for absorption, a point made by Pincus et al., 2015 and evident in the studies of hydrologic sensitivity (De Angelis et al, 2015, and Fildier and Collins, 2015). We'll emphasize the shortwave with a phrase in the revisions.

*6(17-21) I slightly lost the plot here – firstly I am not sure what evidence there is that cloud optical properties is "likely to have a larger impact" is (especially in the context of global mean forcing), and if it is the case, it seems to undermine the reason for focusing on clear skies.* Our point is that differences in the cloud distributions among models – where and when the clouds are – will almost certainly have a larger impact than model errors in treating those clouds in the radiation scheme. It's the errors we're

*6(25) "obscured" – I think the intermodel spread is pretty evident in Collins et al. (2006) where the tropopause standard deviation is almost as big as the mean; I think what has happened in more recent years is a realisation that this really matters for hydrologic(al)*

[Figure]

*sensitivity.*

The narrow range of conditions have obscured important parameterization errors in **sensitivity**, like the sensitivity of solar absorption to water vapor. One can perhaps see this point in retrospect in comparisons like Collins et al. 2006 but its hard to distinguish, say, bias from incorrect sensitivity with a single atmosphere. We'll express this point more clearly in the revisions.

*7(2) I had a few comments on this table. One is that specification of "HFC", rather than a specific HFC, seems too vague to me . . . Second the +4K experiment is well motivated, but not really elaborated on – I guess the focus is on the temperature dependence of transmittance, but this experiment might also have some Planck function dependencies . . . note a typo for the ozone experiment, where it says O2 not O3.*

The atmospheric conditions specify the HFCs precisely. Our current draft specifies CFC11, CFC12, HCFC22 although this list is under discussion.

The +4K experiment is indeed aimed at the temperature dependence of transmittance. It's true that we won't be able to tease apart the Planck function dependence although we're not aware of any evidence that this is a source of error.

We'll add more detail to this table and the surrounding discussion. Thanks for catching the typo as well.

*7-23: Not entirely sure why LBLRTM is singled out here . . .*

LBLRTM is singled out because this is the model with which we, meaning the RFMIP participants, will make reference radiative transfer calculations. Several other groups have agreed to do so informally but we can't commit on their behalf. We will clarify this in the revision.

*7-30 Perhaps the most major of my comments. I naturally assumed that the shortwave GHG forcing would be included for the greenhouse gases, but I realised at this point that while the insolation conditions are specified for aerosols, they are not specified for*

*greenhouse gases . . . it would be advantageous to have a proper day/global average even if it is for a single day, as in the aerosol case. Could the authors at least clarify the situation regarding the GHG SW forcing?*

As Johannes Quaas (reviewer 2) also noted, this table is a little sparse by way of explanation. Solar zenith angle is indeed specified in the experimental protocol, and varies among profiles so that one can compute the diurnal average.

*8-3 "parameterization error increases model diversity" – isn't parameterization error a (and possibly the major) CAUSE of model diversity (unless all models have the same error!)* As above, RFMIP-IRF will assess the error in clear-sky and aerosol radiative transfer parameterizations. We can call this "error" because the correct answer is known for radiation. The same is not true for clouds or convection or a host of other parameterizations for which there are reference calculations analogous to line-by-line calculations.

*8-9 This feels like sloganeering, and I haven't really got a clue what "the 20th century belonged to sulphate" means (there are many contenders to the accolade of "the 20th century belonged to ..."! I might vote for the Beatles), especially given the claim that a strong negative aerosol forcing is implausible and the statement that greenhouse gas forcing dominates at 2-23. Incidentally, is the reference to Carslaw et al., in the following sentence, the right one? There seems little or nothing on sulphate global optical depths in the cited paper.*

*8-13 "Starting in the mid-1970s" – maybe better "Since the mid-1970's"? I presume the stated changes (five-fold, factor of two etc) refer to the present day compared to mid-1970s?*

We have rewritten this paragraph to make it less playful. The reference to Carslaw et al. was to their 'Extended Data Table 2' which gives SO2 emissions, so the original statement assumed (consistent with the literature) that the the conversion of SO2 to sulfate and the lifetime of SO2 did not change dramatically. To avoid confusion we

have made these statements more precise.

"In the 20th Century sulfate is thought to have contributed substantially to the net radiative forcing, although how is disputed (Stevens 2015). What is to disputed is that precursor $SO_2$ emissions increased greatly, and that these emissions were concentrated over a relatively small portion of the planet. Consistent with other studies, Carslaw et al. (2013) estimate that $SO_2$ emissions, to which the dominant component of the aerosol contribution to ERF are attributed, increased three-fold through the first hundred years of industrialization. Smith et al., (2011) pinpoint these changes to changes in the North Atlantic sector – a region covering about a tenth of Earth's surface. Beginning in the 1970s air quality controls began to reduce emissions in Western Europe and North America. Present Western European emissions are now estimated to be a fifth, and North American Emissions a half, of what they were in the early 1970s. As emissions over the Atlantic sector declined, emissions over South and East Asia increased so that globally anthropogenic SO2 emissions remained roughly constant. The short life-time of sulfate implies that the regional concentration of emissions would lead to commensurately large regional forcing. So to the extent that sulfate forcing is important globally, regional signals should be readily identifiable, and may help bound the overall radiative forcing attributable to anthropogenic $SO_2$ emissions."

*8-29 "less negative" – isn't this "more negative"?*

Yes thank you.

*9-3 typo "response"*

Thanks for catching the typo, but in reviewing this sentence we also found it unwieldy. We have revised it to read:

By more tightly-constraining the pattern of the aerosol effective radiative forcing across models it should be easier to identify a clear response of the climate system to the imposed aerosol perturbations. To the extent that clear responses can be identified,

they may be combined with formal methods of detection and attribution (e.g., Stott et al., 2010) to also estimate the magnitude of the forcing. 9-18 This section was written in a different way to the equivalent in Section 3. I could not see a reference to Table 5, and the section is a bit cluttered by the acronyms of the experiments, which are a bit opaque (to me). Perhaps this section could be restructured (not a big job) to do as was done in Section 3, and leave the acronyms to the table, but ensure the underlying motivation of experiments is spelled out.

*9-31 "warming" – sorry if I am getting confused, but as I understand, RFMIP only provides evidence of "forcing" rather than "warming". Perhaps it is the coupling with DAMIP that is being referred to here?*

Here we did mean warming, as we will have the temperature record from the historical simulations and we will compare these directly. To make this more clear we have modified the sentence to say: "The historical simulations based on MACv2-SP will be analyzed, also in cooperation with DAMIP, to test the hypothesis by Stevens (2015) that the observed northern-hemispheric warming is inconsistent with an aerosol radiative forcing more negative than about -1W/m$^2$"

*10-2 "warming hole" –same comment as above. In fact the whole of this sentence, seems to go a bit beyond what RFMIP results could achieve (although they give important clues), so I wondered if this was really referring to DAMIP.*

Yes, we understand how this might be confusing. We have revised the sentence to make clear that simply the identification of a pattern of warming that is consistent with that which is often attributed to aerosol forcing would provide a constraint on that forcing. The revised sentence reads:"For example the pattern, or lack thereof, of the response across the multi-model ensemble may be helpful to advancing our understanding of the extent to which aerosol forcing underlies the warming hole in the east-central United States . . . "

---

## Short Comment (SC1) · 8 Jul 2016

Thank you to the authors for having put this paper together. I have the following comments:

More important comments:

- it would be of benefit to the entire RFMIP/CMIP6 community to have an explicit, complete list of variables to provide for RFMIP in this RFMIP description paper. This list should be further divided into Tier 1 and other tier variables if need be.

- as already mentioned by J. Quaas, and to emphasize further on the first above comment, do RFMIP require on-line diagnostics of the components of the forcing? and if

yes, what are the recommendations for these diagnostics?

- it is somehow disturbing to have a detailed description of simulations under 4.1, while this is not the case for the other two aspects/questions of RFMIP.

Other comments:

p9 : l 23 : There is no reference to MACv2-SP in the Eyring et al, 2015 GMD paper

p10 "piClim-anthro simulation described in Table 1"

p12 Table 1 Title: I would suggest to take out "with interactive vegetation" as some (many?) climate models do not implement this feature

p12 Table 1: this is no difference in the description of RFMIP-ERF-GHG and RFMIP-ERF-LU

p12 Table 1: RFMIP-ERF-AerO3x01 : the description should be : Changes in RFMIPERF- Aer03 scaled by 0.1

p12 Table 1: RFMIP-ERF-AerO3x2 : the description should be : Changes in RFMIPERF- Aer03 scaled by 2

page 13 Table 2: same comment as above, I would suggest to take out "with interactive vegetation"

page 13 Table 2: RFMIP-ERF-HistAer should not the CMIP6 label be : piClim-histaer?

page 13 Table 3: Title 1st sentence: should end "error in forcing".

page 14 Table 5: Experiment titles are in the form of '*SpAerO3*' while in the text titles are in the form of '*SpAer*': is this ok?

---

## Short Comment (SC2) · 13 Jul 2016

The CMIP Panel is undertaking a review of the CMIP6 GMD special issue papers to ensure a level of consistency among the invited contributions, also in answering the key questions that were outlined in our request to submit a paper to all co-chairs of CMIP6-Endorsed MIPs. We very much welcome the important contribution from the RFMIP to CMIP6, and below are a few comments:

General comments:

1. The relationship to AerChemMIP is mentioned in passing a couple of times (e.g. P. 7, line 12; P. 9, line 24), but perhaps a bit more can be said of the connections between

[Figure]

RFMIP and AerChemMIP. For example, will both use the same method for computing radiative forcing? Will there be coordination regarding aerosols?

2. There is the impression in the community that it is too difficult to compute radiative forcing because the traditional definition was for net radiation at top of troposphere with stratospheric adjustment. In RFMIP it is mentioned more than once (e.g. P. 1, line 19; P. 5, line 11; P. 5, line 31) that ERF is now simply the net radiative imbalance at the top of atmosphere (unless we're misinterpreting something). If this is a correct impression, this indeed makes computing ERF much easier, and it may be worth noting this as a significant new aspect of comparing radiative forcing among models. If this is an incorrect impression, it would be worth clarification.

Specific comments:

1. Please update "Eyring, 2015" to "Eyring, 2016" (e.g. p. 2, line 8; p. 4, line 23; p. 9, line 23; p. 11, line 12): Eyring, V., Bony, S., Meehl, G. A., Senior, C. A., Stevens, B., Stouffer, R. J., and Taylor, K. E.: Overview of the Coupled Model Intercomparison Project Phase 6 (CMIP6) experimental design and organization, Geosci. Model Dev., 9, 1937-1958, doi:10.5194/gmd-9-1937-2016, 2016.

2. P. 9, line 27: Readers may be a bit confused by the aerosol protocols. Do the tier 1 aerosol-only experiments allow both prognostic and concentration-driven formulations that may exist in the various models? Understandably the desire for use of a common aerosol concentration data set (MACv2-SP) is spelled out, but what if groups use prognostic aerosols and want to run aerosol-only experiments? Do they not then participate in RFMIP?

3. P. 11, line 2: The warming hole has been shown to have subsided after about 2000, with evidence given to support the idea of remotely-forced atmospheric circulation-driven processes being mostly responsible (Meehl, G.A., J.M. Arblaster, and C.T.Y. Chung, 2015: Disappearance of the southeast U.S. "warming hole" with the late-1990s transition of the Interdecadal Pacific Oscillation. Geophys. Res. Lett., 42, 5564-5570,

doi:10.1002/2015GL064586.). This could be mentioned to contrast to possible aerosol-driven processes.

4. P. 11, line 4: Another recent paper that could be mentioned here is Smith et al (2016, Role of volcanic and anthropogenic aerosols in the recent global surface warming slowdown. Nature Clim. Change, doi:10.1038/NCLIMATE3058) who attempt to provide evidence that the IPO in the Pacific is aerosol-forced (the Pacific analog to Booth et al for the Atlantic).

5. Table 1: the experiment "RFMIP-ERF-LU" has "present-day greenhouse gases" in the description column, but shouldn't it be "present-day land use"?

With many thanks for your ongoing efforts in the CMIP6 process.

The CMIP Panel

---

## Author Comment (AC2) · 23 Aug 2016

Dear Gerry -

Thanks for providing the CMIP panel's perspective on our manuscript. Some points below.

*The relationship to AerChemMIP is mentioned in passing a couple of times ... but perhaps a bit more can be said of the connections between RFMIP and AerChemMIP. ...*

We agree and have added more information. They will compute forcing the same way in most instances and have a complimentary approach to aerosols. We can't find the

text you reference on page 7 but assume you meant page 5, which we've added to:

"The complimentary Aerosols Chemistry Model Intercomparison Project (AerChem-MIP, Collins et al. 2016) ERF simulations adopts the same radiative forcing calculation methodology as RFMIP for Tier 1 experiments. AerChemMIP deliberately targets inter-active chemistry models and extends RFMIP to allow the community to further decompose present day aerosol and non-CO2 forcings into a larger set of forcings caused by different sets of precursor emissions."

*There is the impression in the community that it is too difficult to compute radiative forcing because the traditional definition was for net radiation at top of troposphere with stratospheric adjustment. In RFMIP it is mentioned more than once (e.g. P. 1, line 19; P. 5, line 11; P. 5, line 31) that ERF is now simply the net radiative imbalance at the top of atmosphere (unless we're misinterpreting something). If this is a correct impression, this indeed makes computing ERF much easier, and it may be worth noting this as a significant new aspect of comparing radiative forcing among models. If this is an incorrect impression, it would be worth clarification.*

Thanks for pointing this out. We've amplified this point on on page 1:

"Rapid adjustments are generalizations of (and replace) the stratospheric adjustment (Hansen et al., 1997) that has historically been used to account for the impact of rapid stratospheric equilibration on top-of-atmosphere radiation fluxes. Accurate diagnosis of ERF requires custom model integrations . . . The diagnosis of ERF from such simulations is simplified, however, because the ERF is diagnosed from changes in top-of-atmosphere radiation."

*Please update "Eyring, 2015" to "Eyring, 2016"*

We have updated the citation.

*2. P. 9, line 27: Readers may be a bit confused by the aerosol protocols. Do the tier 1 aerosol-only experiments allow both prognostic and concentration-driven formulations*

*that may exist in the various models? Understandably the desire for use of a common aerosol concentration data set (MACv2-SP) is spelled out, but what if groups use prognostic aerosols and want to run aerosol-only experiments? Do they not then participate in RFMIP?*

Groups must indeed use the MACv2-SP aerosol formulation to participate in the SpAer component of RFMIP, though they may participate in the other components with any aerosol description. We have clarified the text to make this point more clearly.

*3. P. 11, line 2: The warming hole has been shown to have subsided after about 2000, with evidence given to support the idea of remotely-forced atmospheric circulation-driven processes being mostly responsible . . .*

We decided not to cite this paper as it diminishes the motivation for participating in RFMIP-SpAer.

*4. P. 11, line 4: Another recent paper that could be mentioned here is Smith et al (2016, . . . doi:10.1038/NCLIMATE3058)*

We added this citation.

*5. Table 1: the experiment "RFMIP-ERF-LU" has "present-day greenhouse gases" in the description column, but shouldn't it be "present-day land use"?*

Ah, yes, thanks for catching that.

---

## Author Comment (AC3) · 24 Aug 2016

Dear Dr. Michou -

Thank you for providing these helpful comments.

**More important comments**

*it would be of benefit to the entire RFMIP/CMIP6 community to have an explicit, complete list of variables to provide for RFMIP in this RFMIP description paper. This list should be further divided into Tier 1 and other tier variables if need be.*

The full description of the requested data fields is intimately tied up with the formal

CMIP data request.We hesitate to provide so much detail when the information is available elsewhere, in a form we can update as needed. It also seems imprudent to allow for conflicting requests in the manuscript and the database. We looked at some of the other manuscripts in this special collection and adopted similar language, namely:

"We provide brief summaries of requested output but the definitive and detailed specification is documented in the CMIP6 data request available at https://earthsystemcog.org/projects/wip/CMIP6DataRequest.''

We also amplified the data request for the ERF integrations:

"The main diagnostics are the top of atmosphere energy budget terms required to estimate ERF. Diagnostics of atmospheric state, including temperature, water vapor, cloud and aerosol information, are requested to allow for detailed diagnosis of rapid adjustments. A few daily variables related to temperature and precipitation are requested in conjunction with DAMIP to help distinguish direct effects of external forcing and air-sea interaction effects on historical changes in extreme indices (e.g., extreme precipitation)."

*as already mentioned by J. Quaas, and to emphasize further on the first above comment, do RFMIP require on-line diagnostics of the components of the forcing? and if yes, what are the recommendations for these diagnostics?*

We have clarified that on-line diagnostics are not part of the protocol but may be provided:

"We are also interested in comparing IRP and cloud adjustments estimated from the kernel method with those that have been explicitly calculated in models that employ the triple radiation call approach of Ghan (2013) to diagnose instantaneous forcings and cloud adjustments. As this method is time-consuming and not implemented by all models we do not include this request as part of the protocol but models implementing triple radiation calls are encouraged to contact us. ''
*it is somehow disturbing to have a detailed description of simulations under 4.1, while this is not the case for the other two aspects/questions of RFMIP.*

As part of other revisions we expanded the description of the IRF calculations in section 3, after which we read the manuscript with this point in mind. Our sense is that this imbalance reflects the different levels of maturity of the ERF calculations described in section 2 and the specified-aerosol simulations in section 4. At this stage it seems important to express the latter in detail but needlessly long to add information to the former (indeed, there's already quite a lot of motivating detail about the use of fixed-SST runs and the use of climatology).

We are open to further input if this point seems especially important.

**Other comments**

*p9 : l 23 : There is no reference to MACv2-SP in the Eyring et al, 2015 GMD paper*

We have changed the reference to the MACv2-SP description to the now-available Stevens et al., 2016.

We have omitted the ERF simulations for aerosols.

We have clarified in section 2 that the treatment of vegetation is to follow each model's normal use, i.e.

"Land-surface models including interactive vegetation, if available, should be applied as in normal integrations."

We have harmonized the experiment titles and experiment_ids to be self-consistent. Aerosols and ozone are prescribed together and denoted AerO3. Experiments seeking to diagnose radiative forcing have titles beginning "RFMIP-ERF" and experiment_ids denoting the protocol ("piClim"). We have also been careful to use the same descriptions throughout the manuscript.

Thanks again for your careful review.

---

## Author Comment (AC4) · 7 Sep 2016

Dear Johannes -

Thanks very much for your detailed and helpful suggestions. We've followed nearly all of them. We have corrected all the typographic errors you found but don't bother listing them below.

**Major comments**

*The paper only occasionally indicates which model diagnostics (output) is to be produced and analysed. One could only assume that the different contributions of top-of-atmosphere (also surface?) radiation flux densities (solar and terrestrial, all-sky and*

[Figure]

*clear-sky) are to be determined. What about cloud quantities?*

Consistent with other MIP manuscripts we note at the end of section 1:

"We provide brief summaries of requested output but the definitive and detailed specification is documented in the CMIP6 data request available at https://earthsystemcog.org/projects/wip/CMIP6DataRequest."

We have also added further detail. For the ERF simulations the most important diagnostics are top of atmosphere energy budget changes. These are simple to produce and we want to encourage as many groups to produce these and participate. Surface energy budget data and cloud data allow much more interesting analyses. In the protocols we ask for ISCCP cloud simulator data, but not all groups can provide this, so we do not want to sound as if they are vital in this overview paper. These diagnostics are now mentioned explicitly. We have added text here at the start to be more explicit:

"The main diagnostics are the top of atmosphere energy budget terms required to estimate ERF. Diagnostics of atmospheric state, including temperature, water vapor, cloud and aerosol information, are requested to allow for detailed diagnosis of rapid adjustments. A few daily variables related to temperature and precipitation are requested in conjunction with DAMIP to help distinguish direct effects of external forcing and air-sea interaction effects on historical changes in extreme indices (e.g., extreme precipitation)."

*When it comes to the aerosol ERF it has been shown that it is useful to on-line diagnose components of the forcing. This in particular involves a triple-call to the radiation, allowing to diagnose the radiative forcing due to aerosol-radiation- and due to aerosol-cloud interactions (Ghan, Atmos Chem Phys 2013; doi:10.5194/acp-13-9971-2013). Would it not be useful to request such a diagnostics?*

We agree this is useful. However, as not all models have this capability and as there is no clear protocol for how to employ the methodology beyond greenhouse gas and

aerosol changes we have chosen not to make it part of the official request. We have added text asking for these calculations if they are available.

"We are also interested in comparing IRP and cloud adjustments estimated from the kernel method with those that have been explicitly calculated in models that employ the triple radiation call approach of Ghan (2013) to diagnose instantaneous forcings and cloud adjustments. As this method is time-consuming and not implemented by all models we do not include this request as part of the protocol but models implementing triple radiation calls are encouraged to contact us. "

*For the transient ERF, more specifications are necessary how to compute it. Are multiple ensemble members necessary? Or are time slices computed? Or else is a noisy signal accepted? What about the strong deviation from the pre-industrial base state at least in the future scenarios at least in the Arctic?*

We agree and have added details. We also refer to the Forster et al. 2016 study which expands in more detail. We also add more details on the ERF time slice error for consistency. The modified text reads

"Transient simulations (Table 2) in which forcing agent concentrations evolve over time are designed to give a complete picture of the CMIP6 Historical transient ERF and possible future radiative forcing. Transient ERFs will be computed by differencing top of atmosphere energy diagnostics from three ensemble members employing time varying forcing changes with the energy budget diagnostics from the Tier 1 30-year control simulation. These integrations will use the same prescribed preindustrial climatology of SST and-sea-ice as in the time-slice ERF experiments. AerChemMIP employs a more complex method of prescribing SSTs and sea-ice that allows for base climate changes through time. Offline tests found that such complexity was unnecessary as ERF was only weakly dependent of base-state with small differences in the future confined to sea-ice edges (Forster et al. 2016). Therefore RFMIP adopts the same base climatology in all experiments for ease of implementation. Tests also found that the transient

[Figure]

ERF fields suffer from year-to-year random noise, so ten-year averages of the three ensembles would be needed to quantify ERF to within 0.05 W/m2 (Forster et al. 2016)"

*For the IRF study, the paper should be structured more clearly to clarify the two aspects (CO2 and aerosols) to this, which use two very distinct approaches.*

We have followed the suggestion below of isolating each of these efforts in its own subsection.

*For the imposed aerosol forcing, it would be good to indicate what the implied ERF due to aerosol-radiation-interactions and what the ERF due to aerosol-cloud-interactions are for e.g. year 2011.*

As described below we have provided more detail on the MACv2-SP climatology, including that the all-sky ERF for 2005 is -0.7 W/m2 in one CMIP6 model.

**Minor comments**

*page 1*

*l19 the "roughly" might merit a sentence of explanation, or a reference.*

We have changed this phrasing:

"...can induce a radiative perturbation loosely called a *radiative forcing*"

*is it worth mentioning that Eq. (1) actually defines the radiative forcing, and that the usefulness of F is linked to the degree to which α is independent on the exact nature of the process that generates F, so that comparing different F is sufficient to predict different ΔT for a given α (i.e. given model)?*

We considered changes along these lines but decided against them as we could not find a way to express these points without distracting readers away from our focus on the radiative forcing itself.

*Page 2*

*L20: If sticking to the acronym "IRP" this should probably be spelled out "instantaneous radiative flux perturbation" (assuming the "R" represents "radiative")*

We have removed the use of the IRP acronym. In a few places we use the acronym IRF for instantaneous radiative forcing, as per the fifth IPCC report, but in general we have tried to distinguish carefully between forcing (i.e. a change in physical or chemical state); radiative perturbations including IRF; and effective radiative forcing.

*l30: this sentence in my opinion is not understandable to somebody not experienced in the topic. The point is about efficacy, and so I'd propose to write "...suggests that IRP due to different forcing agents is not, in practice, a very good predictor for changes in surface temperature assuming constant climate feedback parameters, a point..."*

We have revised this text in a way we hope will be more clear without bringing in the concept of efficacy.

"Equation 1 is a diagnostic framework, useful in interpreting observations and comprehensive models of the climate system. Experience with models (in which all terms can be determined precisely) suggests that IRF is not, in practice, related very closely to changes in surface temperature, a point highlighted . . . "

*Page 3*

*l2 "accurate diagnosis", or rather diagnosis at all?*

We chose not to change this text as forcing can also be diagnosed from historical and 1%/year runs following Taylor and Forster, 2006, doi:10.1175/JCLI3974.1.

*L5 Reference Hansen et al. (2005): I think it would be appropriate to also cite Rotstayn and Penner (J Climate 2012, 14, 2960-2975) who introduced the concept several years before Hansen*

Reference has been added.

*l6 "fullness of the model response" to be precise, perhaps "fullness of the rapid model*

*response"*

We chose not to change this text as timescale is not the essential difference between feedbacks and adjustments.

*page 4: l2 "concentration or emission changes", since at least some models computed carbon- and aerosol cycles interactively*

Agreed, and changed accordingly.

*l15 One would expect a reference corroborating the sentence, rather than – once more – introducing the regression concept (Gregory).*

We've amplified this point:

"The "fixed-SST" method has important advantages compared to regressions of top-of-atmosphere imbalance against surface temperature change (**?**). The first is better error characteristics (Forster et al. 2016): thirty years of simulation using only the atmospheric and land components of an earth system model can diagnose global ERF to better than 0.05 W/m-2 standard error, such that a 2xCO2 forcing of 3.7 W/m-2 is larger than its standard error over 70% of the globe. Achieving similarly small errors from regression requires ensembles of coupled model integrations and therefore many centuries of simulation."

*L19 The sentence is not straightforward to understand. For both methods, only a given perturbation can be diagnosed, and this can be done for either approach. L22 It would be useful to specify as clearly as possible the simulation. I'd suggest to clarify the following things: (i) it is one annual cycle consisting of 12 geographical distributions of sea surface temperatures and sea ice fractional coverage; (ii) from how many years should the fields be derived? Average over any 30 years of the piControl run? (iii) should the monthly means be linearly interpolated to the individual time step, or abruptly change on the first of each month, 0 UTC?*

We've added some detail:

"The protocol for RFMIP fixed-SST integrations is to use a monthly-averaged model-specific climatology of SST and sea- ice based on the model's preindustrial DECK integration (Eyring et al. 2016). Applying a climatology limits variability and improves the diagnoses of small ERF differences. The same climatology will be used for all ERF integrations. We request that distributions from a monthly averaged climatology of SST and sea-ice fractional coverage covering the annual cycle be generated from at least a 30 year segment of a preindustrial control integration. These should be prescribed according to the AMIP protocols, whereby interpolated daily data is generated preserving the prescribed monthly averaged fields. Because ERF is weakly dependent on background state (Forster et al. 2016) the exact choice of background SST and sea-ice has little impact on the forcing estimate in the historic period and has only a small effect in future climates (see below). We hope that a simple approach will encourage model centers to participate.

Time-slice simulations (Table 2), in which forcing agents are held constant at present-day or 4xCO2 values, provide estimates of present-day and 4xCO2 ERF. Present-day estimates provide a direct comparison between the estimates of ERF in the model with other estimates e.g. in assessment reports (Myhre et al. 2103). Estimate of ERF will also let us understand basic aspects of each model's temperature and other climate responses in the Historical and 4xCO2 DECK simulations. ''

*L26 "without compromising accuracy" - I don't understand what is meant in this context.*

Text now deleted as paragraph reworded.

*L27 "present-day" needs clarification. Is this year 2011 as in CMIP5, or 2005 as in CMIP3? It would also be good this time to have the date consistent with the end of the DECK historical simulation.* and also your points *Table 1 "Present-day" is not specified. Is this intended?* and *RFMIP-ERF-GHG Experiment description: It should be made very clear whether ozone (tropospheric? Stratospheric?) is considered a*

*(greenhouse) "gas". Can one not be specific by writing "CO2, CH4, N2O, Halocarbons, and O3 and CO precursor gases?*

We have clarified that RFMIP follows CMIP6 protocols that define these quantities:

"RFMIP follows CMIP6 protocols, so that *present-day* is interpreted as 2015 and *greenhouse gases* refer to those specified by Meinshausen et al. (2016), i.e. CO2, CH4, N2O, and some or all of a long list of halocarbons or equivalent concentrations. Ozone concentrations are specified separately. "

*The Tier-2 experiments need to be motivated. Is a signal from 0.1 Aer really detectable over the noise in a 30-year fixed-SST framework? Why a factor of 0.1 and not a factor of 0.5? Or why not – probably even more useful in interpreting the historical simulations – a year 1985 simulation?*

We have omitted these experiments from the protocol.

*Page 5:*

*l2 Forster et al. (the cited JGR paper) say 30 years of integration time are necessary to characterize ERF. How is this done here? By 30 ensemble members and they reporting the forcing every year of the simulation? Or is a much greater transient uncertainty simply accepted if a single ensemble is run?*

Text now added to better describe transient ERF estimates – see additions above

*It seems Forster et al. (JGR) claim the base state does not matter much so a pre-industrial SST and SIC distribution is good enough. But is this not a main uncertainty when integrating into the future? How meaningful is a forcing diagnostics in the second half of the 21st century in the Arctic when pre-industrial sea ice cover is prescribed?*

We show in Forster et al, submitted JGR that this is in fact only a small effect around sea-ice edges. Text has been clarified and expanded (see above)

*Page 6*

[Figure]

*l14: ''both greenhouse gases and aerosols'': It is quite unclear from Table 3 how the greenhouse gases beyond CO2, and aerosols are to be prescribed, in both the control and perturbed simulations. Only at some point in the text, it appears the atmospheric profiles should be not only cloud-free, but also aerosol-free. I didn't find any information about greenhouse gases beyond CO2.*

This has been addressed by clarifying that RFMIP follows the CMIP6 protocols, as above.

*Page 7*

*l1 Why not provide the selected profiles as supplementary material? It would be useful to specify in the paper the exact way the profiles are introduced. Is the vertical discretisation the model's one? Or is a common vertical grid chosen?* and also *L8 "when finalized" why not put this up for review here as well?*

We have amplified the description of the selected profiles including specifying that all radiation models are meant to solve the same problem i.e. operate on the same vertical grid.

"Modeling centers are asked to use the vertical grid provided."

As the column selection is somewhat involved full details will be described elsewhere; our intent here is to describe the protocol.

*L4 "aerosol-free": What about greenhouse gases?*

We couldn't figure out how to make this more clear. The columns are chosen to optimize a forcing calculation for greenhouse gases so indeed those gases (at both pre-industrial and present-day concentrations) are part of the calculation on which the column selection is based.

*L12: It would be good to make clear at the beginning of section 3 that there are two distinct and very different approaches to characterize greenhouse gas- and aerosol*

*forcing. It would be also useful to split these two into to sub-sections, one on greenhouse gases, and one on aerosols.*

We've followed this suggestion.

*L14: "radiative perturbations": does this imply a double radiation call, one with the current aerosol and a second one with all aerosol set to zero? The exact definition should be clarified, and also the necessary output. Is this requested for the top of the atmosphere, or vertically resolved? Broadband in the solar spectrum, or spectrally resolved? Why not both, the clear-sky and the clean-sky flux?*

We've substantially expanded the discussion of the protocol in this section:

"RFMIP is developing a compact (roughly 100) sample of atmospheric conditions (profiles of pressure, temperature, humidity, greenhouse gas concentrations, surface properties) and radiative transfer boundary conditions (solar geometry and solar constant) that, when weighted appropriately, can be used to estimate time-averaged global-mean fluxes . . . Present-day atmospheric and surface conditions are sampled from reanalysis while greenhouse gas concentrations follow the CMIP6 protocol, using 2015 values provided by Meinshausen et al. 2016. Aerosols are not included. Perturbations to these states allow for the calculation of instantaneous radiative perturbation as the difference in flux between a perturbed state and present-day conditions and concentrations. Some perturbed states (see Table 3) represent changes in conditions tied to CMIP DECK or Historical simulations. The more idealized perturbations described in Table 4 are aimed at exposing model errors with global impacts, especially in present-day forcing by specific greenhouse gases. This set of conditions will be distributed on the Earth System Grid as a single file.

The sample is constructed to minimize the sampling error in annual-mean, present-day clear-sky, aerosol-free forcing by greenhouse gases (i.e. the difference in fluxes using present-day and pre-industrial gas concentrations). The sampling error, even with as few as 50 distinct conditions, is several orders of magnitude smaller than the

forcing; forcing errors for other composition changes are larger but still small relative to the change in flux. Further details on the selection of these columns will be reported separately.

Modeling centers are asked to compute broadband (spectrally-integrated) fluxes for the full range of conditions and all perturbations using off-line versions of their radiative transfer parameterizations (or using any work flow that computes fluxes as the host model does using precisely the specified conditions). Modeling centers are asked to use the vertical grid provided and to omit aerosols. The representation of greenhouse gases, and particularly the choice of using a subset of gases or one of the equivalent concentrations provided by Meinshausen et al. 2016, should follow used in other integrations made for CMIP6 and related activities. ''

*l17: It would be good to be exhaustive in the request list, i.e. to also include which other parameters (temperature, pressure, specific humidity? gases?) to provide.*

We have slightly expanded the list and also note that the full specification provided in the file is complete.

*L24: Maybe would it be possible to write why multiple reference models are advantageous? Is it expected that their results differ?*

We decided not to address this point because it is a bit subtle but mostly irrelevant to the protocol. We did amplify the role of the reference models:

"We anticipate that reference models may also be used to assess the impact of choices made in the CMIP6 specification for greenhouse gas concentrations (Meinshausen et al., 2016) including the use of equivalent concentrations to reduce the number of greenhouse gases considered, the neglect of species like CO that are not well mixed, and the specification of latitudinal and vertically-varying concentrations for well-mixed gases. ''

*Page 8:*

*l6 Another substantial contributor is the diversity in simulated cloud distributions (Penner et al., Atmos Chem Phys 2006 doi:10.5194/acp-6-3391-2006; Stier et al., Atmos Chem Phys 2013, doi:10.5194/acp-13-3245-2013)*

We have added these citations after new text: "further modulated by varying distributions of clouds"

*l9: 20th century: rather a balance of sulfate and greenhouse gases; L12: To be more precise: the continents adjacent to the North Atlantic; l17: "commensurately large", or why "larger" (than the emission changes?)*

This paragraph has been substantially revised.

"In the 20th Century sulfate is thought to have contributed substantially to the net radiative forcing, although the magnitude and mechanisms are disputed (Stevens,2015). What is not disputed is that precursor SO2 emissions increased greatly, and that these emissions were concentrated over a relatively small portion of the planet. Consistent with other studies, Carslaw et al. 2013 estimate that SO2 emissions, to which the dominant component of the aerosol contribution to ERF are attributed, increased three-fold through the first hundred years of industrialization. Smith et al., 2011 pinpoint these changes to changes in the North Atlantic sector – a region covering about a tenth of Earth's surface. Beginning in the 1970s air quality controls began to reduce emissions in Western Europe and North America. Present Western European emissions are now estimated to be a fifth, and North American emissions a half, of what they were in the early 1970s. As emissions over the Atlantic sector declined, emissions over South and East Asia increased so that globally anthropogenic SO2 emissions remained roughly constant. The short life-time of sulfate implies that the regional concentration of emissions would lead to strong regionality in forcing. So, to the extent that sulfate forcing is important globally, regional signals should be readily identifiable and may help bound the overall radiative forcing attributable to anthropogenic SO2 emissions."

*l26: "reducing temperature" may be misunderstood (the GHG effect mostly is domi-*

*nant), why not – pertinent to the paper – say that they introduce a negative effective radiative forcing?*

We have qualified the sentence to read "reducing global mean temperature," but we wish to stress the ultimate impact of aerosols on climate (temperature) rather than the intermediate step of radiative forcing.

*l23 Why citing Eyring here?*

The reference for MACv2-SP is now available and has replaced this reference.

*L23: The sentence in brackets is difficult to understand and should be explained better. Page 10*

We have omitted the sentence.

*l33: But does the MACv2-SP provide a forcing more negative than -1 Wm-2 in 2011?*

No, and we have amplified this point:

"In a version of the Max Planck Institute Earth System Model (MPI-ESM; see Giorgetta et al., 2013; Stevens et al. 2013) using an updated atmospheric component the clear-sky ERF for this aerosol description is -0.8 W/m2, evaluated over the time period 2000-2011 using aerosols at 2005 values. The corresponding all-sky ERF is -0.685 W/m2. In other models the latter value, especially, will depend on the model's distribution of cloudiness. ''

*l21: "quite useful" for what?*

From the context we think this point is clear: output from the ISCCP simulator will be useful in studying cloud adjustments. RFMIP's use of the data is described in detail in section 2.2, referred to in the sentence.

*CMIP6 label/experiment id: can the dashes be omitted? I think having just words helps in some scripts, and this was common practice in CMIP5.*

We have not received this feedback from the CMIP or WGCM infrastructure panels. As our names are quite long we will retain the dashes at this time.

*RFMIP-ERF-AerO3: Experiment description: How is O3 perturbed for models that include atmospheric chemistry, i.e. that require emission, rather than concentration, perturbations? Is this tropospheric ozone only?*

Text has been added to better describe transient ERF estimates – see additions above

*RFMIP-ERF-LU Experiment description: gases or land-use?*

This was an error: we mean changes in vegetation types and corresponding changes in surface albedo, roughness length and transpiration but not changes in gases.

*RFMIP-ERF-4xCO2: Does this require another control simulation for models with an interactive carbon cycle (namely a simulation with pre-industrial CO2 concentrations, rather than emissions)?*

Interactive carbon cycle models are not used here

*RFMIP-ERF-HistNat Experiment description: I think "etc." is very bad to use in such a protocol. It is necessary to very precisely say what should be varied. What could and should be thought of besides solar activity and volcanic eruptions?*

Yes. We removed etc. and specify that variability is to include any spectral variation

*RFMIP-ERF-HistAer Experiment description: What about ozone?*

The title and experiment_id have been changed to indicate that ozone and aerosols are always treated together.

---

## Author Response (AR1)

**The Radiative Forcing Model Intercomparison Project**

Robert Pincus, Piers M Forster, and Bjorn Stevens

**Response to reviews**

We are grateful to the reviewers and others offering comments for their efforts in helping us clarify the protocol for the Radiative Forcing Model Intercomparison Project. We have provided line-by-line responses to all of the reviewer comments in response to each reviewer. To summarize the major changes:

> We have distinguished carefully between forcing, meaning a change in physical or chemical state; radiative forcing, meaning a resulting change in the radiation field; and effective radiative forcing ERF, meaning the temperature response multiplied by the system's feedback parameter. We have removed the "instantaneous radiative perturbation" language in the initial submission and retained a few instances of the well-defined "instantaneous radiative forcing."

> We have noted that RFMIP follows the CMIP6 experimental protocol and explained several ramifications of this choice.

> We have clarified the requested output for ERF simulations, noting that full details are available in the formal CMIP6 request. We have elaborated further on choices related to these simulations such as the use of climatological boundary conditions, and explained that results from multiple radiation calls are welcome but not required.

> We have amplified the description of the conditions to be provided and the data requested for the assessment of parameterization error.

> We have added more detailed description and a new reference for the aerosol climatology used in the specified aerosol forcing activity.

> Experiment names and _ids have been made self-consistent. Two ERF experiments related to aerosols have been removed and a few of the greenhouse gas offline experiments have been revised.

We believe the resulting manuscript contains enough detail that modeling centers will be motivated and fully capable of participating in the experiments.

A version of the manuscript marked to show changes follows.

[revised manuscript text omitted]

sations with T. Andrews, <s>W. D. Collins, D.</s> D. R. Feldman, E. J. Mlawer, L. Oreopoulos, and D. Paynter<s>,</s>. W. D. Collins and V. Ramaswamy originated the idea of assessing parameterization error in treatment of aerosols (section 3.1.2). We thank Daniel R. Feldman of Lawrence Berkeley National Lab for carefully assembling the novel data request for the assessment of aerosol radiative forcing.

**Table 2.** Experiments for diagnosing time-evolving effective radiative forcing. Three-member ensembles of atmosphere-only integrations interactive vegetation and using sea-surface temperatures and sea ice concentrations fixed at model-specific pre-industrial control climatology. Forcing  post-2015 uses a scenario consistent with DCPP and DAMIP (SSP2-4.5)

| Experiment Title | CMIP6 Label (experiment_id) | Experiment Description | Start | End | Major Purposes |
|---|---|---|---|---|---|
| RFMIP-ERF-HistAll | piClim-histall | Time-varying forcing from all agents. | 1850 | 2100 | Diagnose transient ERF from |
| RFMIP-ERF-HistNat | piClim-histnat | Time-varying  ERF from volcanoes, solar  (including spectral) variability | 1850 | 2100 | Diagnose transient natural ER |
| RFMIP-ERF-HistGHG | piClim-histghg | Time-varying  ERF by greenhouse gases | 1850 | 2100 | Diagnose transient ERF fro house gases |
|  RFMIP-ERF-HistAerO3 | piClim-histaerO3 | Time-varying  ERF by aerosols | 1850 | 2100 | Diagnose transient ERF from |

**Table 3.** Sets of atmospheric conditions to be supplied by RFMIP for assessing parameterization error in  clear-sky top-of-atmosphere flux changes. The entire set of conditions is described as CMIP Experiment RFMIP-IRF with CMIP6 Label (experiment_id) rad-irf

| Atmospheric conditions | Gas concentrations | Major Purpose | Relevant Experiment |
|---|---|---|---|
| Present-day | Present-day | Baseline | |
| Present-day | Pre-industrial | Present-day radiative forcing | Historical |
| Present-day | $4\times$ pre-industrial $CO_2$ |  Radiative forcing from $4\times CO_2$ | abrupt4xCO2 |
| Present-day | "future" |  Radiative forcing in future conditions | RCP8.5 at 2100 |

**Table 4.** Sets of atmospheric conditions to be supplied by RFMIP for assessing radiative forcing by specific agents and probing sources of parameterization error. The entire set of conditions is described as CMIP Experiment RFMIP-IRF with CMIP6 Label (experiment_id) rad-irf

| Atmospheric conditions | Gas concentrations | Major purpose |
|---|---|---|
| Present-day | Pre-industrial $CO_2 \times 0.50$ |  Radiative f |
| Present-day | Pre-industrial $CO_2 \times 2$ |  Radiative f |
| Present-day | Pre-industrial $CO_2 \times 3$ |  Radiative f |
| Present-day | Pre-industrial $CO_2 \times 8$ |  Radiative f |
| Present-day | Pre-industrial $CO_2$ | Present-day radiativ |
| Present-day | Pre-industrial $CH_4$ | Present-day radiativ |
| Present-day | Pre-industrial $N_2O$ | Present-day radiativ |
| Present-day | Pre-industrial  $O_3$ | Present-day radiativ |
| Present-day | Pre-industrial HFC | Present-day radiativ |
| Present-day +4K | Present-day | Assess error in tem |
| Present-day +4K |  Present-day, water vapor increased to present-day relative humidity | Assess error in sens |
| Pre-industrial | Pre-industrial | Sensitivity of comb concentration/cond |
| "Future" | "Future" | Sensitivity of comb concentration/cond |
| Present-day | Last glacial maximum per PMIP | Assess errors under as per Kageyama e |

**Table 5.** RFMIP simulations with specified anthropogenic aerosols (SpAer). All simulations are all based on the MACv2-SP prescription of anthropogenic aerosol optical and cloud active properties. They are only to be performed to replicate other simulations in the DECK, within the ERF component of RFMIP, or within the Detection and Attribution Model Intercomparison Project (DAMIP) in the case when MACv2-SP is not used as the default aerosol climatology in the parent simulation.

[revised manuscript text omitted]

---

## Author Response (AR2)

**The Radiative Forcing Model Intercomparison Project**

Robert Pincus, Piers M Forster, and Bjorn Stevens

**Response to the topical editor**

We are grateful for the topical editor's quick assessment of our revisions and for his very careful reading of our manuscript. We have corrected the typographic errors. We have also changed the language at the beginning of section 4 to be clear that RFMIP-IRF assess both parameterization weakness (in the treatment of gases) and, indeed, errors (in the treatment of aerosols).